# Syntalos: a software for precise synchronization of simultaneous multi-modal data acquisition and closed-loop interventions

Matthias Klumpp [1], Lee Embray[1], Filippo Heimburg[1], Ana Luiza Alves Dias[2], Justus Simon [1], Alexander Groh [1,3], Andreas Draguhn [1,3] & Martin Both [1,3] ✉

Complex experimental protocols often require multi-modal data acquisition with precisely aligned timing, as well as state- and behavior-dependent interventions. Tailored solutions are mostly restricted to individual experimental setups and lack flexibility and interoperability. We present an open-source, Linux-based integrated software solution, called 'Syntalos', for simultaneous acquisition and synchronization of data from an arbitrary number of sources, including multi-channel electrophysiological recordings and different live imaging devices, as well as closed-loop, real-time interventions with different actuators. Precisely matching timestamps for all inputs are ensured by continuous statistical analysis and correction of individual devices' timestamps. New data sources can be integrated with minimal programming skills. Data is stored in a comprehensively structured format to facilitate pooling or sharing data between different laboratories. Syntalos enables precisely synchronized multi-modal recordings as well as closed-loop interventions for multiple experimental approaches. Preliminary neuroscientific experiments on mice with different research questions show the successful performance and easy-to-learn structure of the software suite.

Many experiments in modern neuroscience aim at linking different system levels (e.g., cellular, network and behavioral) and modalities (e.g., electrophysiological and imaging data)[1–3]. Such parallel recordings make it necessary to integrate data from different acquisition systems[4,5]. A vital requirement for correct analysis and interpretation of results is the precise alignment of these heterogeneous data, often at a millisecond time scale[6]. However, data integration and synchronization has remained a major challenge. Several programs have been developed as generic tools for data acquisition and instrument control (e.g., LabVIEW) or as more specialized tools for behavioral experiments (e.g., ANY-Maze or Noldus EthoVision XT, and Bonsai as an open-source program; see Supplementary Table S1). However, given the increasing complexity and diversity of experimental designs, the need for a general-purpose, easy-to-use and open-source solution remains. At present, precise synchronization of heterogeneous data streams remains a major problem, especially for long observation periods and if no external synchronization signals can be used. Key requirements for a precise and versatile data acquisition system are: (1) integration of multi-modal data from behavioral, imaging and electrophysiological measurements; (2) consistent and accurate timing of in- and outputs over prolonged periods; (3) versatile and user-friendly implementation of new data sources or actuators; (4) storage of data in

¹Institute of Physiology and Pathophysiology, Medical Faculty, Heidelberg University, Heidelberg, Germany. ²Brain Institute, Federal University of Rio Grande do Norte, Natal, Brazil. ³Interdisciplinary Center for Neurosciences (IZN), Heidelberg University, Heidelberg, Germany. ✉e-mail: mboth@physiologie.uni-heidelberg.de

a widely usable format and comprehensive file structure for open access. At present, these properties are difficult to achieve with existing tools for users without advanced programming skills.

We therefore developed a highly versatile system for parallel data acquisition and processing which meets the requirements of multimodal experimental settings. A key feature of the system, called 'Syntalos', is that timestamps from all data sources are aligned to one globally shared Master clock. Timestamp divergences between individual devices are continuously detected and corrected for, ensuring precise synchronization over several hours. Analysis and data interoperability are facilitated by a directory structure combining data from different sources with their metadata and using the same formats for similar types of data. These features are implemented into the core of the software such that they are shared between all data sources or online data processing functions, facilitating implementation of new modules via intuitive application programming interfaces (APIs). For features requiring high computing power and very precise timing/low latency, a C++ API exists, while items with less tight requirements can be implemented via an easy-to-use Python API. A wide range of commonly used input sources have already been implemented into the default version of Syntalos, including a module for electrophysiological recordings (Intan RHX, Intan Technologies, Los Angeles, California), in vivo-calcium imaging (UCLA Miniscope, Los Angeles, California) and various video cameras (see methods section). Closed-loop interventions are supported by an Arduino input/output interface which is custom-programmable in Python and yields latencies of 2–6 ms. For lower latencies in the hundred-microseconds range, Syntalos also has a user-friendly way to execute MicroPython code on dedicated microcontrollers. Furthermore, most pre-existing Python-based algorithms or analysis tools can be implemented as additional modules with little to no code changes, running in their own, isolated virtual environments. Here we outline the general architecture and properties of Syntalos, and we demonstrate its efficacy and reliable timing in a typical experimental setting. Syntalos provides a versatile solution to the serious problem of data synchronization and integration, and it facilitates exchange of data and analytical methods between laboratories.

## Results

A major challenge in neurophysiological studies is to synchronize individual recording components (e.g. camera and electrophysiology recordings). Accurate and continuous signal alignment is a prerequisite for establishing correct relationships between neurophysiological and behavioral data. To systematically explore the impact of correct signal alignment on the interpretation of neurophysiological data we recorded time-sensitive neurophysiological data in a sensory discrimination experiment, using Syntalos (Fig. 1A). In this experiment, neuronal spike patterns are recorded in the somatosensory cortex while mice sample apertures of varying widths (narrow, wide) with their whiskers, which are recorded with a high-speed camera. Spikes and whisker touches were originally optimally aligned by Syntalos. We then arbitrarily de-synchronized both signals, simulating a systematic time-shift between both recording devices during the course of the experiment. We then asked how this temporal misalignment affects the prediction of stimuli (slit width) from the simultaneously recorded spike trains (Fig. 1B).

If the alignment between high-speed video images and electrophysiological signals is incrementally shifted by 1 ms per second (leading to -1 s time shift over the course of a 960 s long experiment), the prediction accuracy of the classifier drops from nearly 100% to near chance level (Fig. 1B). At the same time, the number of units whose firing behavior correlates with the width of the aperture drops from 65% to 17%. This example highlights the importance of continuously synchronizing devices to avoid cumulative timing errors. In many cases, synchronization is done exclusively by flashing LEDs at the

beginning of the recording, but then not updated to not disturb the animal. In other cases, devices are only activated for short periods during the course of the ongoing experiment, again requiring regularly updated synchronization for each recording period. This is the case in the present example (Fig. 1A, B) where the high-speed cameras are only activated for short periods to keep the amount of stored data low (Fig. 1C). In summary, correlative measurements of physiological and behavioral parameters require precise temporal alignment of recording devices throughout the whole experiment. Syntalos supports such continuous updating of time for multiple recording devices or actuators, as shown in the present experiment (Fig. 1A–D)[4,7].

### General architecture

In Syntalos, different devices (e.g., amplifiers, cameras, gates, food dispensers etc.) are conceptualized as distinct 'modules' with input and output interfaces. A module can be thought of as a physical device (e.g., an amplifier) that processes a data stream and then sends it elsewhere. Modules are connected using virtual 'wires' (Supplementary Fig. S1A). This abstraction allows flexible support of any experimental design in software, using tailored modules for data acquisition, processing, storage or operation of actuators. Typically, each apparatus or online analysis process will correspond to a single module, but modules may also be created as combinations of multiple processing steps for the convenience of the user.

With this, Syntalos provides a framework which runs and coordinates individual modules (Fig. 2). The modules are dedicated to defined work packages, such as acquiring, processing or storing data or interacting with the behavioral setup. The Syntalos engine provides a declarative API (application programming interface) for the modules to interface with the program. Modules can be written in C/C++ or Python. Support for further programming languages (e.g. C#, Rust or Java) can be added via Syntalos' out-of-process module interface, as long as the language in question supports interfacing with C. Several complex tasks (synchronization, threading, data storage etc.) are handled by the Syntalos engine, such that modules only need to implement code for their dedicated purpose instead of dealing with low-level complexity. Besides being user-friendly, this design enforces uniform handling of recurring tasks such as data storage or timestamp handling between modules.

Using C/C++ for modules allows Syntalos to run most operations at native speed, without any overhead of an interpreted scripting language or a just-in-time (JIT) compiler[8]. Fast reaction times are also supported by highly parallel processing through multi-threading, making use of the multiple cores available on modern CPUs[9]. Notably, thread execution time is organized by the Linux operating system scheduler which has been optimized to give tasks their fair share of CPU time[10–12]. In addition, Syntalos can advise the scheduler to mark tasks for preferential processing, or assign specific CPU affinities for latency reduction. The scheduler also ensures that one highly demanding task cannot starve other tasks for CPU resources. Critical procedures (e.g. data acquisition; Fig. 2A "Module A") can be assigned to dedicated threads at a higher priority, such that they run with the least amount of interruptions. Other, less time-critical procedures are variably assigned to threads which can be shared between different tasks, using an event-based system to call individual subroutines (Fig. 2A, "Module B", "Module C"). Thread assignment is done by the Syntalos engine based on preferences set in the module's code (available at https://github.com/syntalos/syntalos).

As an alternative to C/C++, modules may also be written in Python, a widely used high-level programming language[13]. This allows simple integration of new modules by a wide range of users, reusing existing code or creating basic scripts in Python. Such additional modules run outside of the main Syntalos process, using a dedicated Python virtual environment without interfering with other modules or with the main application. However, in such cases data must be serialized to be sent

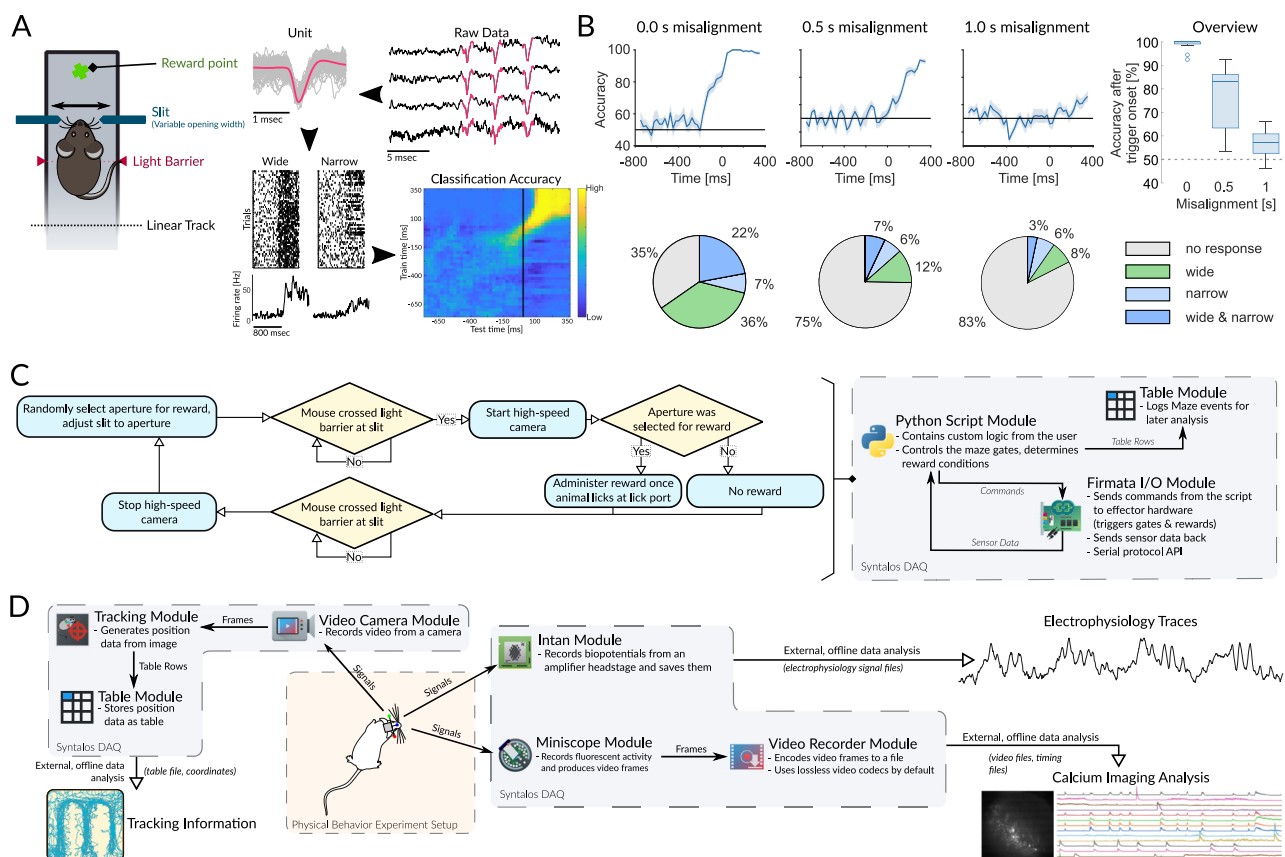

**Fig. 1 | Example experiment and modular design of Syntalos. A** Behavioral task, in which mice touch apertures of varying widths with their whiskers. Whisker-aperture interactions are recorded with a camera and electrophysiological data is recorded with a chronic tetrode array. Single unit spike patterns are identified offline. Spiking is aligned to the whisker touch and a classifier is trained with the spiking of several single units to predict the animal's behavior. **B** If whisker touch and spikes are misaligned by systematic incremental shifts of 1 ms per second (leading to 0.5 s (middle panel) or 1 s (right panel) over the course of a 16 min long experimental session), the prediction accuracy drops to near chance levels. Additionally, the number of units whose firing correlates with whisker touch drops from 65% to 17% (pie charts, lower panels). Error bands of decoding accuracies indicate the standard error of the mean (SEM). Box plots represent the decoding accuracy in the time window ranging from trigger onset to 400 ms after trigger onset, with the respective values for each condition as median [lower quartile, upper quartile; whisker minimum, whisker maximum]: No time shift: 100 [99.12 100; 98.5 100]; 0.5 s time shift: 83 [63.38 86.12; 53.5 92.5]; 1 s time shift: 57 [52.62 60.88; 46 66]. Data is shown for 413 barrel cortex units (pooled from 6 animals). **C** Flowchart

illustrating the logic of the behavioral-physiological experiment. This logic can be programmed in Python by the experimenter within the Syntalos Python Script module. The Python module can receive data from and send commands to the Firmata I/O module, which is an implementation of the Firmata serial interface API to ultimately control an Arduino board which reads the actual data from sensors and commands any connected effectors based on the rules programmed in the Python script. **D** Schematic representation of a complex example experiment. During the experiment, Syntalos will acquire a multitude of data, and perform some online analysis. The animal is recorded via a camera while it traverses a maze or any other behavior setting. The video recording is handled by a Syntalos camera module, which produces frames that are analyzed by a tracking module for their tracking information which is finally saved by the table module for later offline analysis. Electrophysiological data can be acquired by a module dedicated to the Intan hardware, while a Miniscope pipeline can be set up using the Miniscope module and a Video Recorder module to store the acquired frames for later calcium activity analysis. Icons/logos reproduced with permission (see Acknowledgements).

to the external Python adapter process, and subsequently de-serialized for use by the Python process. The same applies in reverse for data sent to Syntalos by a Python module (Fig. 2A, see "Module D"). These additional steps, while autonomously organized by Syntalos, slow down Python-written modules slightly and increase latencies[14,15] (see below). To increase robustness and ensure that a misbehaving module will not crash the whole recording, the method of running as separate process is also available to modules written in other programming languages via the *libsyntalos-mlink* shared library.

Modules have 'ports' to receive data input and to provide output data to other modules. Ports are visualized on an intuitive GUI, and inputs and outputs of modules can be connected by the user by drawing lines between ports of the same type (Supplementary Fig. S1). Each port can only handle one specified data type, e.g. image frames or table columns. Syntalos supports several pre-defined data types (see Supplementary Table S1). New types can be added by modifying the Syntalos engine, which requires understanding of Syntalos' C++ codebase.

During operation, the same data may be used by different modules, running in different threads. In order to safely pass data between threads and to avoid read/write conflicts, a lock-free queue implementation with a single writer and a single consumer is used[16]. When a module creates data, it will write them into the queues of all connected modules, which then use these data once they need it (Fig. 2A; data streams between threads). This organization avoids additional memory allocations in non-Python modules. Once a data block is generated by a module, it is considered immutable and must not be modified in-place again. Thus, other modules can read from, but not write to, the respective data block. An output port of a module is internally represented by a 'data stream'. A module can request data from a given output by connecting to it via a process called 'subscribing'. Many modules can subscribe to the same data stream, but one module input port can only be subscribed to one stream at a time (streams can not be merged). During an experiment run, the Syntalos engine continuously monitors all module connections. A user warning is generated if a

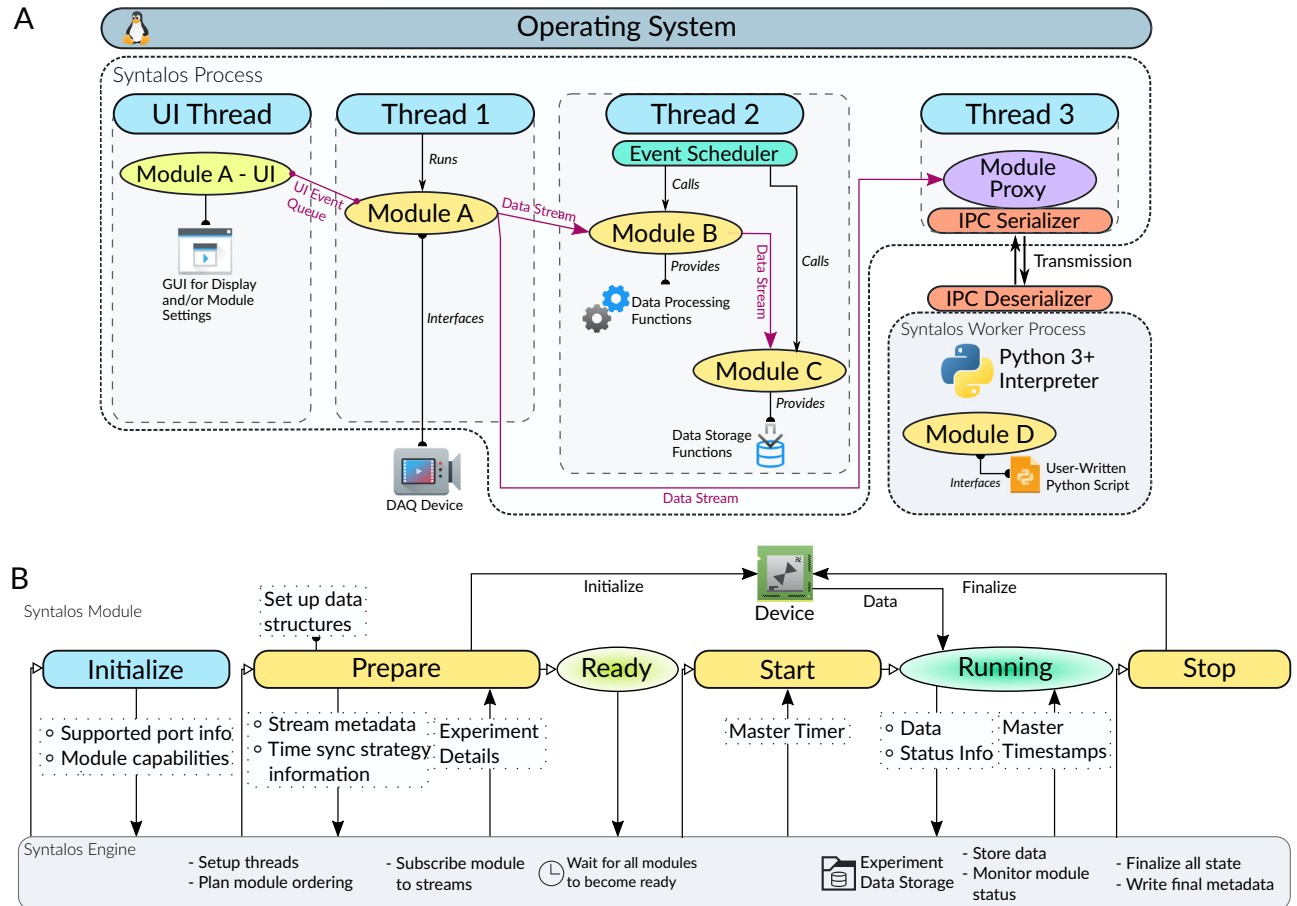

**Fig. 2 | Architecture of the Syntalos engine and module run cycle. A** Shows how Syntalos manages parallelism with threads between modules. Module A runs in part in the UI thread to display for example a settings panel, and in part in a Syntalos worker thread (Thread 1) to acquire data from a DAQ device. This data is sent via a data stream to Module B, which is combined together with Module C in a different worker thread (Thread 2). Those two modules share the time provided by the given thread and are called by the event loop of Thread 2 at selected intervals. They also use a data stream to move data between two modules, as the stream itself does not care whether the modules live in the same thread. The DAQ data is also streamed to Module D, for which Thread 3 serializes communication for IPC (inter-process communication) with the actual Module D, which is written in Python and executed as a separate child process outside of the main Syntalos process. **B** Shows the states

a module goes through during an experiment run, from left to right. The upper part ("Syntalos Module") depicts the actions the module itself executes and the states it is in, while the lower part ("Syntalos Engine") depicts the tasks Syntalos itself performs. The vertical arrows show communication between the two sides. Initially, when the module is created ("Initialize" phase) it submits basic information (which ports it supports etc.) to the engine. When the user runs an experiment, all modules set up their required data structures and devices in a prepare step, signal readiness to the engine and then are started at once. At that point they will also get access to the master timer, to acquire master timestamps. During the run, they will store experiment data and are monitored by the engine, until the experiment is eventually stopped and the modules finalizes its data and possibly state of a device that it manages. Icons/logos reproduced with permission (see Acknowledgements).

module cannot read data fast enough so that data queues between modules are getting too large and the computer is in danger of running out of memory.

**Program behavior during experiment**

When a new module is added to an experiment configuration, it communicates a set of basic information to the Syntalos engine, such as the ports which it supports, or threading requirements. Once the experiment is started, the module runs through a series of states (Fig. 2B): First, the Syntalos engine sets up the threading model and basic data structures, then an execution order for all modules is planned, such that modules generating data are started before modules consuming these data. Modules will then be triggered to enter the first 'prepare' stage in which they initialize any hardware, communicate their time synchronization strategy and set runtime-specific metadata on data streams (such as the expected bit-depth of a camera recording or expected headers of a CSV table). When the module is ready, it will transition into the 'ready' state and communicate this information to the engine. The engine will subscribe all input ports to data streams according to the user's configuration. Once all modules have signaled

that they are 'ready', the engine will emit a 'start' signal, and modules transition into a 'running' phase where they start to process or emit data. The Master clock is only available once the 'start' signal has been sent. This strategy ensures that modules start almost simultaneously, within their technical limitations. The engine continuously monitors the state of all running modules and data queues, and provides modules with Master clock timestamps (see below). Once the user stops the experiment, the engine emits a 'stop' signal and triggers modules to finish their data processing and data acquisition tasks. Remaining data left in queues will still be processed for a few seconds. If a module fails (e.g., if a device is disconnected) during an ongoing measurement, all data will be saved and the experiment run is stopped. However, modules can be exempted from this 'stop on failure' behavior in the GUI. Once all modules are 'stopped', the Syntalos engine will finalize the EDL (Experiment Directory Layout) metadata and clean up system resources which have been used for the data acquisition run (Fig. 2B). For details on EDL and comparison with other data formats see Supplemental Information "Data Storage & Formats" and https://edl.readthedocs.io/latest/intro.html). With these features, Syntalos provides a user-friendly framework for integration of multiple modules

and secures precise synchronization of all active processes during an experiment.

## Time synchronization

To ensure synchrony between data-acquiring modules, their timestamps are aligned to a single clock. Syntalos used Linux' `CLOCK_MONOTONIC` clock type, which is usually based on the system's CPU timestamp counter and has microsecond precision[17]. This Master clock is considered to be accurate by definition. All secondary clocks, for example those from data acquisition systems, will be compared to the Master clock to determine and correct for their offset (Fig. 3A). Within Syntalos, any data blocks are always paired with their respective master timestamps, so that all data processing modules receive accurate time information for any incoming dataset.

To maintain synchrony, Syntalos provides modules with *Synchronizer* constructs, which quickly quantify and correct the offset of a secondary clock to the Master clock. Syntalos contains two different synchronizers, one for devices providing timestamps and one for devices providing a continuous data counter with known frequency. Modules can use different strategies to react to time divergences: i) write the offset information to a special *tsync* binary file, ii) correct the received timestamp to match the master time, iii) adjust the clock / data acquisition (DAQ) speed of the external device, iv) use a combination of these strategies (Fig. 3A). Which method is chosen depends on whether the external device supports clock adjustments, and whether the retrieved data will be stored in a format supporting error-corrected timestamps. If the storage format does not support adjusted timestamps, a *tsync* time synchronization file is written, which enables simple re-synchronization of timestamps during offline data processing. Some devices provide neither a timestamp nor a data counter. For strictly polled instruments, Syntalos then takes the mean of the Master clock's time before request for data and response as timestamp. For devices with buffers, Syntalos calculates the apparent recording time backwards, using the time of data receival and the fill state of the buffer.

The buffers of DAQ devices may confer delays based on hardware properties, limited speed of data transmission or varying activities of the operating system[12,18,19]. Thus, the time points when data is read from the DAQ buffer may not accurately reflect the time of data generation. Therefore, internal timestamps of all DAQ devices are continuously compared with the Master clock. To robustly correct time divergences between the Master clock and a device clock, Syntalos calculates the moving mean and variance of differences between master and device clock timestamps. The number of time points used for calculating mean and variance are set for each module within the code provided by the module author, either as a fixed value or as a dynamically adjusted parameter, e.g. depending on sampling frequency. If no value has been specified, the number is defined by a formula, which has secured good synchrony between devices during test runs:

$$n = \left\lceil \left( f + \frac{1}{0.01 + \left(\frac{f}{250}\right)^2} \right) *10 \right\rceil \text{ (f = sampling frequency)}$$

Syntalos' time synchronization algorithm consists of two phases, initialization and continuous updating. When an experiment is started, the differences between the timestamps of the peripheral device and the Master clock are calculated. From this, the median (with an added, module-defined tolerance range) and variance (with a fixed tolerance range) are obtained. These values are used as reference for synchronization during the subsequent continuous updating phase. In this phase, mean and variance are calculated for each sliding window, and compared to the reference values. Outliers will increase the standard deviation, typically without major alteration of the mean

– in this situation, the device timestamp will be adjusted to match the expected Master clock timestamp. If only the mean value of timestamp differences exceeds the tolerance range, time values are corrected, either by shifting new incoming data timestamps or by recording the time divergence in a *tsync* file. Additionally, devices allowing external reset of their clock can be set to the Master clock by using the respective device API (Fig. 3A). The maximum tolerances set by modules are stored as metadata.

## Evaluation of time synchronization

We quantified the performance of Syntalos by measuring the synchronization of different optical and electrophysiological recording devices: three different industrial cameras from The Imaging Source and Basler, a UCLA Miniscope[20], a standard webcam, an Intan electrophysiology amplifier, a Raspberry Pi Pico running Syntalos-provided MicroPython code and the Arduino open-source electronics platform running Firmata, a generic protocol for communicating with microcontrollers from a host computer. External events were provided as rhythmic TTL pulses from a signal generator (see Methods). These pulses were used to generate an optical signal via an LED and voltage inputs to the Intan, Arduino, or Pi Pico devices (Fig. 3B). Time deviations between the recorded data and the Master clock were measured for a running time of >24 h (Fig. 3C, D). Note that all devices show a time drift of ~750 ms after 24 h due to a systematic difference in timing velocity between the signal generator and the internal Master clock. The relative timing between devices, however, was stable over the course of the experiment. In the experiment, the original timestamps from the Intan system show a significant drift (light green line in Fig. 3C, D) while timing was well aligned after synchronizing the signals by the Master clock (dark green). Together, these data show that heterogeneous devices are reliably synchronized by the Syntalos algorithm, even over long recording periods. At higher time resolution (Fig. 3E) significant timing differences between the different devices become visible, reaching around 40 ms in case of the UVC webcam. These deviations result from the low video image sampling frequency (25 Hz). Notably, however, the mean deviation stays stable, i.e. there is no cumulative timing difference effect (compare left and right panels in Fig. 3E; note the resetting 'jumps').

One potential confound is variation of the Master clock speed due to temperature changes caused by varying computational load or fluctuations in the environment. To assess whether this has an influence on device synchrony, we analyzed the time difference of recorded timestamps to a linear regression of the data shown in Fig. 3D. Indeed, the timestamps of the Master clock showed fluctuations around the linear interpolation (Fig. 4A). In line with low sampling rate, the time differences between different external devices and the Master clock showed considerable variations on an event-to-event base, due to aliasing effects (Fig. 4B). The median deviation, however, was constant, even over long recording periods of >24 h (Fig. 4C). Thus, all devices stay in good synchrony with the Master clock despite fluctuations of the absolute speed.

Next, we looked at the underlying synchronization process with single timestamp resolution. For this, we divided the Master clock time interval between two *tsync* control points by the number of samples acquired from the Intan recording system between these time points. Ideally, this should result in a sampling interval of 50 µs (with sampling rate set to 20 kHz). Indeed, the measured values varied within ±5 ns (corresponding to ±0.01% of the expected sampling interval; Fig. 4D, E). The time interval for correction of external device times by the Master clock has been set to a minimum of 24 s, but may be longer if deviations are smaller than the threshold value described above. Measured correction intervals for the Intan device were mostly 24 s but could reach values of more than 48 s (Fig. 4E, second panel). Together, these data show stable synchronization of devices over long periods of time.

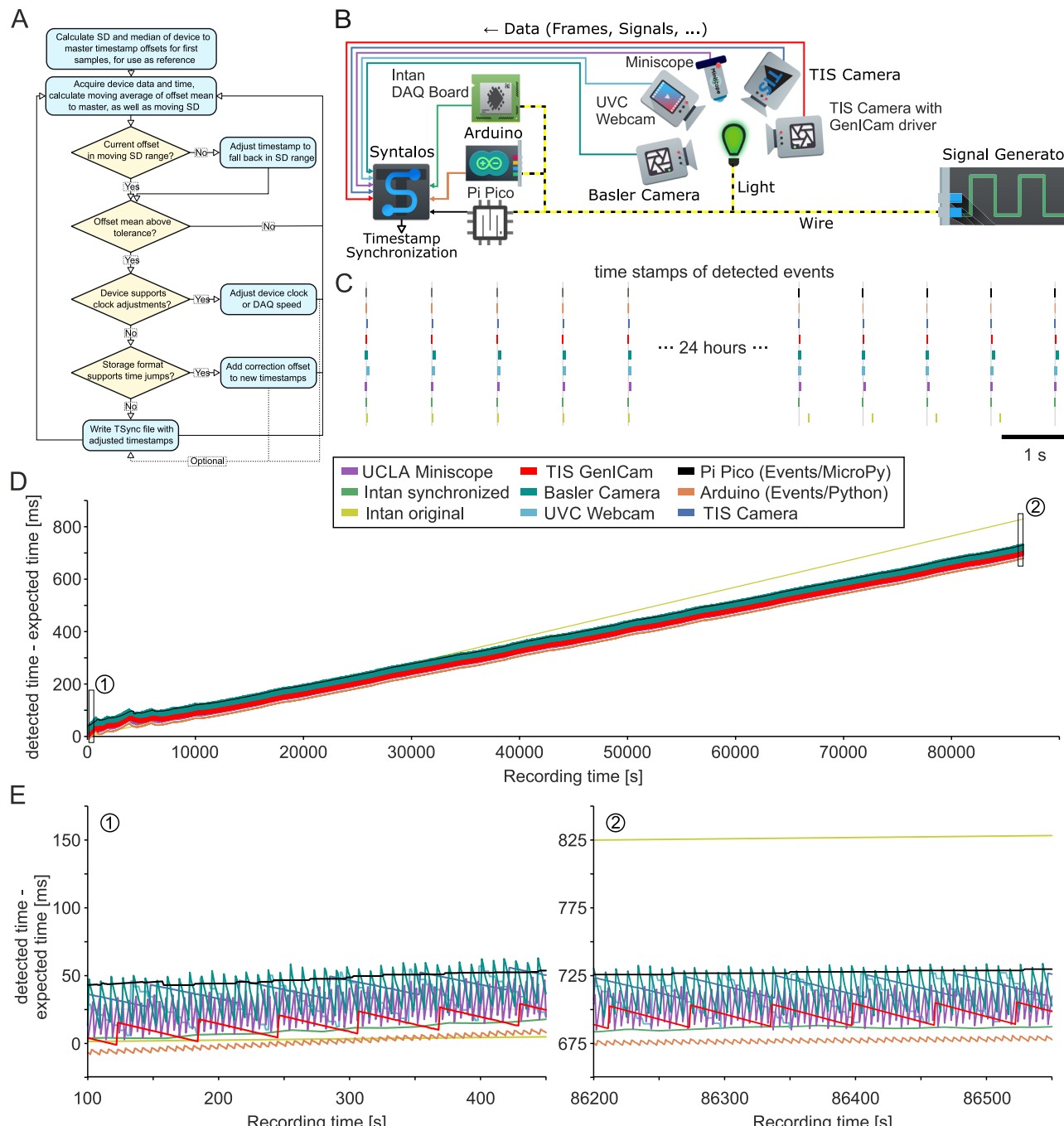

**Fig. 3 | Timestamps assigned by Syntalos for various devices. A** Flow diagram of how Syntalos will adjust new timestamps to fall back in line with previous ones, even if two clocks of separate devices diverge in time [detailed explanation is in results text]. **B** Experimental setup to test the performance and accuracy of synchronization. In this configuration, various devices were used and synchronized by Syntalos: a standard USB UVC (Universal Video Capture) webcam (25 Hz sampling rate), a scientific camera (Basler, 25 Hz sampling rate), two identical scientific cameras (The Imaging Source, 60 Hz sampling rate), one controlled by a dedicated Syntalos module (TIS Camera), the other using the generic GenICam module (TIS GenICam), a UCLA Miniscope (sampling rate 30 Hz), an Intan RHD2000 electrophysiology USB interface board (sampling rate 20 kHz), an Arduino Firmata I/O serial interface connected via USB, and a Raspberry Pi Pico microcontroller connected via USB. The TIS Camera, UVC webcam and the Miniscope have no own independent clocks for timestamping individual frames and are time-synchronized by Syntalos based on driver timestamps. The Arduino and the Pi Pico devices are not time-synchronized in a strict sense but obtains their timestamps directly from Syntalos' own Master clock. The Intan USB interface board is time synchronized but

runs its own internal clock. For the experiment, a signal generator produces 3.3 V positive, 240 ms long square waves every second. This signal is directly fed into the Arduino, the Pi Pico, and Intan digital input port for sampling, while the cameras record an LED connected to the same signal line. **C** Recorded relative time points of the frames (or samples) in which the voltage signal produced by the CED board (or the LED light) is detected. **D** Time deviation of the recorded timestamps from the expected time points. Note that the external signal generator clock of the Intan device is slightly slower than the other clocks which adds up to an error of approximately 150 ms after 24 h of recording. Additionally, the Intan internal clock is also faster than the computer clock. The step-like characteristics of the UVC webcam and the fluctuating signals from the other devices are due to the respective frame rates and aliasing effects. Note that due to temperature fluctuations at the beginning of the recording, the computer clock fluctuates with respect to the Intan clock. **E** Close-up of an early time point (indicated by 1 in panel **D**) and a late time point (indicated by 2 in panel **D**) of the recording. Icons/logos reproduced with permission (see Acknowledgements).

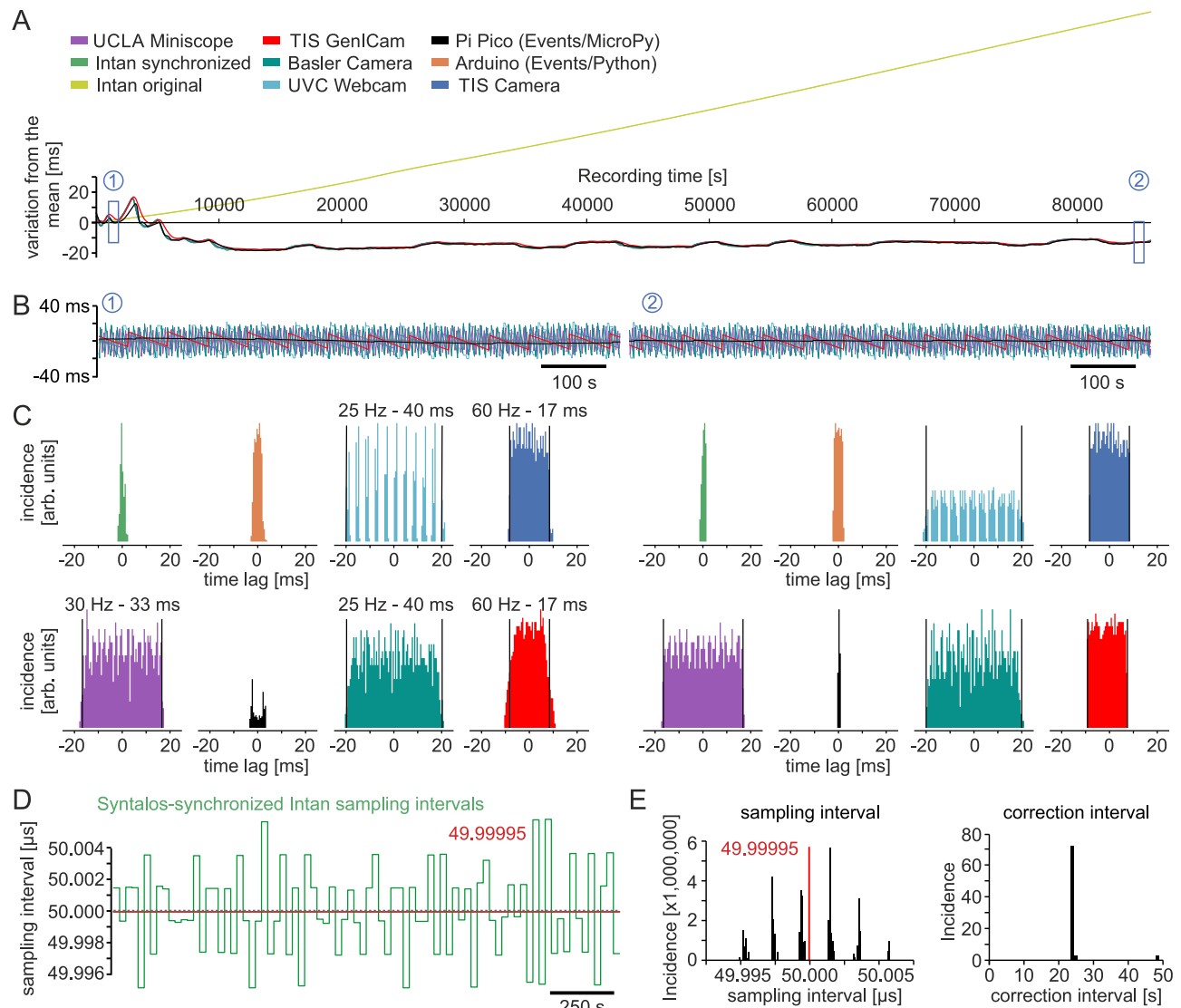

**Fig. 4 | Synchronization performance. A** Fluctuation of the mean detected onset times. Signals in panel D of Fig. 3 were low-pass filtered and deviations from the linear interpolation between the first 10 s to the last 10 s of recordings are displayed. Note that the devices (except from the original, uncorrected Intan data) fluctuate together, indicating that synchronization is accurate even if the internal clock speed varies. This variation is likely due to a combination of temperature shifts and workload differences of the computer. **B** Timestamps of the various devices relative to the low-pass filtered timestamps of the TIS camera (fastest device directly controlled by Syntalos). Left panel is from the time region marked as '1' in panel **A** and right panel from the time region marked as '2'. **C** Distributions of the data depicted in **B**. The timestamps of the devices stay within their expected range i.e. within their frame rate limits. Intan timestamps fluctuate within a full range of 2 ms which are the limits of the Syntalos synchronization margin. Mean value deviation stays well below 1 ms (see Results section). **D** Sampling intervals of the Syntalos-synchronized Intan recordings depicted over recording time. Constant sampling intervals are assigned by Syntalos for periods of ≥24 s based on time differences between the computer clock and timestamps from Intan. Assigned sampling intervals fluctuate around the expected value of 50 μs (i.e. 20 kHz, dashed gray line) by ±5 ns. Mean value is 49.9998 μs (red line), due to slight differences in clock speed between the computer and Intan. **E** Quantification of the corrected sampling intervals. Left panel: distribution of the assigned sampling intervals for the data shown in **D**. Right panel: distribution of the duration of the step-wise correction of sampling intervals. The minimum duration for which constant sampling intervals are assigned is 24 s, a value determined by Syntalos based on the device type and sampling frequency.

Despite the precise internal synchronization, technical limitations of peripheral devices may add constant offsets which are not visible to the clock synchronization algorithm. For example, the moment of data acquisition in a video camera may differ by a constant time lag from the moment of data readout and timestamp assignment (Fig. 3C). Such constant offsets are, by their nature, not visible to Syntalos. Correcting them requires an independent measurement, after which the constant offset value of each device can be subtracted for future measurements. We assessed this systematic error by measuring offsets of six different devices, using the device with the highest sampling rate (the Intan system) as temporal reference (Fig. 5). Again, an external signal was provided by a TTL-triggered LED pulse (1 Hz), and deviations from the time of the light signal were measured for ~15 minutes (Fig. 5A). Indeed, the devices showed considerable offsets of up to 40 ms (Fig. 5B). Individual offsets varied strongly due to aliasing between the devices' sampling rates and the 1 Hz external signal. Mean values of the time lags reflect the systematic error caused by timestamp assignment during data readout. Repeating this experiment for 60 times revealed that this systematic error is constant (within few ms) for each device. Hence, it can be measured and corrected for a specific experimental configuration (Fig. 5B, C).

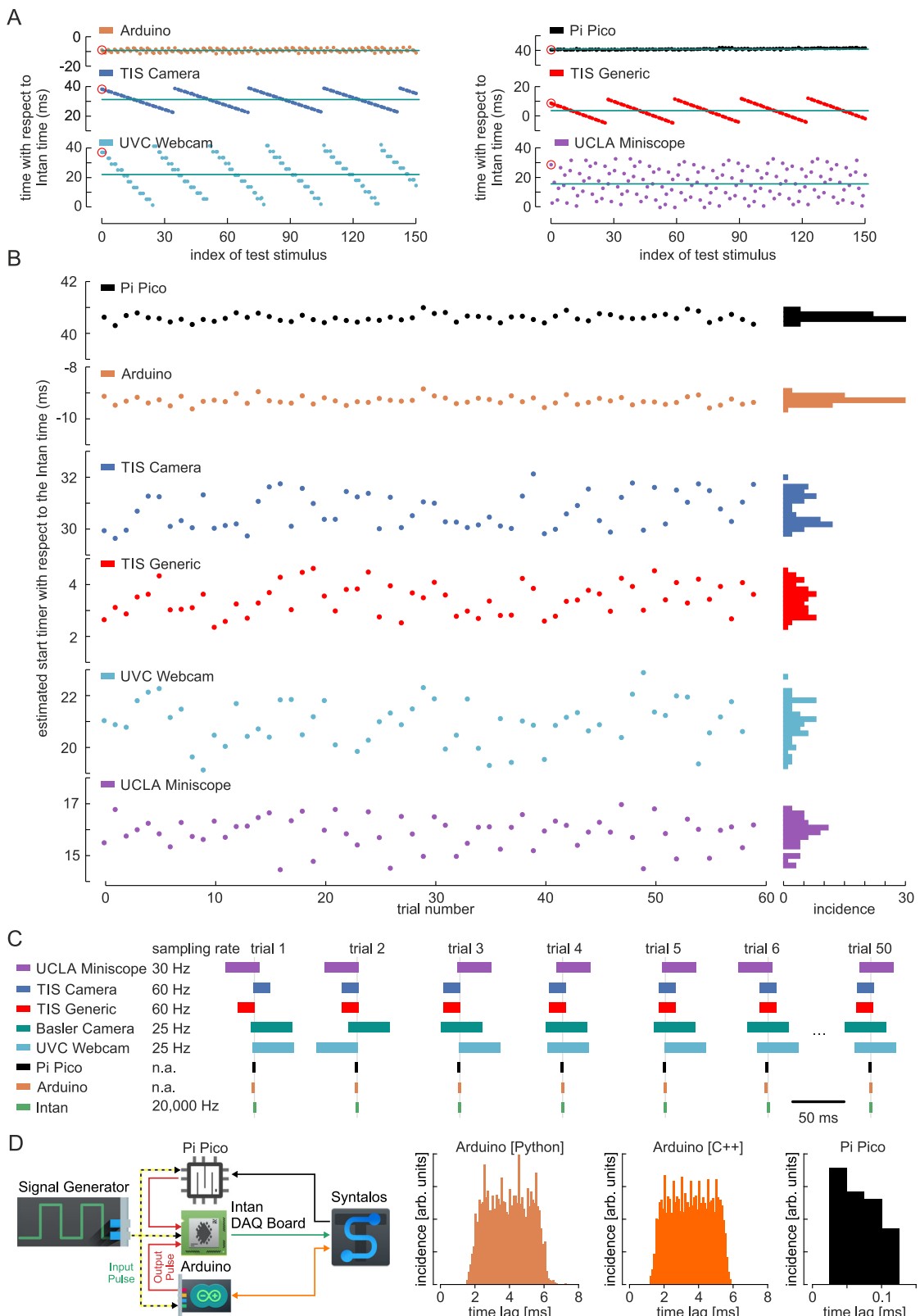

For closed loop experiments, latency is an important measure of how fast the system can react to a certain stimulus. As a test, we devised a simple experiment where a TTL pulse is simultaneously sent to the Intan device and an Arduino Uno. The signal at the Arduino is processed by a Python module within Syntalos, which, as soon as it receives the data, instructs the Arduino to send a new TTL pulse to the Intan device, indicating that the original TTL signal was detected. The

Python module also records the time when it processed data in a table. The time difference of the initial and the Arduino-generated pulse, measured by the Intan device, reflects the roundtrip latency of Syntalos and the Python module. This value represents a realistic, but sub-optimal scenario since a Python module (and therefore an external process) was involved. Modules can also be written in C++ and be loaded into Syntalos as shared library, sidestepping both Python and

**Fig. 5 | Estimated start time of the devices upon multiple starts of a Syntalos recording.** The same Syntalos project configuration is started 60 times and the timestamps are evaluated over 10 min. **A** Example plots for the first 150 stimuli of a synchronization experiment. Times are depicted relative to the Intan time (highest sampling rate). As Syntalos and the independent clock of the signal generator are not synchronized, the first detected event occurs at a random time point (red circle). Sampling rate of the devices are not exact multiples of one second such that the relative detection onset times shift over time and are reset when the difference gets larger than one frame (especially well visible for the TIS camera and the UVC webcam). The exact start time of the device by Syntalos can be estimated by averaging the relative onset times (red line). **B** Scatter plots (left) and histograms (right) of the estimated start times of the devices with respect to the Intan recording (highest sampling rate). The start times of the different devices differ by up to ~40 ms. The jitter of the start times, however, is much smaller (~3 ms) well below the sampling intervals of the devices. **C** Sampling windows of different devices for seven trials. Due to the correction of offsets (shown in **B**) all sampling windows fall onto the correct time of a LED-emitted light signal (thin vertical line in each trial). **D** Roundtrip latencies for closed-loop experiments. Left panel: Schematic representation of the latency test. Signal generator sends an input pulse to the Pi Pico or Arduino, respectively. After detection, an output pulse is generated. The time lag between input- and output pulses is detected via Intan by Syntalos. Right panels: Distribution of the input-output latency for the Pi Pico running MicroPython, the Arduino board controlled by a Python script in Syntalos, and the Arduino board controlled by a module written in C++. Icons/logos reproduced with permission (see Acknowledgements).

inter-process communication (IPC) latencies. For certain scenarios, they can react much faster, but in this case the maximum speed is mainly limited by the Arduino and USB communication itself, instead of by IPC latency and Python (Fig. 5D, middle panel). The average roundtrip latency was 4 ms, with a maximum of about 6 ms (Fig. 5D). This is sufficient for most behavioral experiments. However, for cases where deterministic fast reactions to events with sub-millisecond precision are required, Syntalos can integrate and program microcontrollers running MicroPython. For that, we used a Raspberry Pi Pico, which receives a MicroPython script from Syntalos to perform the same action as the Arduino. During the experiment, Syntalos communicates with the Pi Pico to receive data processing times from it. With this setup, test-runs achieved roundtrip latencies of <150 μs (Fig. 5D, right panel). Alternatively, we recommend specialized hardware and software that can provide real-time guarantees and can be controlled by Syntalos (example experiment using our GALDUR board in Fig. S4, with code available at https://github.com/bothlab/labrstim).

We compared the timing accuracy of Syntalos with that of Bonsai, a widely used open-source system[21,22] in behavioral neurosciences (Supplementary Fig. S5). Several video cameras were kept in sync over several hours by both systems, and closed-loop roundtrip latencies for the Arduino were comparable. However, without external timing cues or additional hardware, Bonsai was not able to synchronize the Intan DAQ-board to the connected video cameras and the UCLA Miniscope (time deviations were larger than 9 s after 22 h of recording). In addition, the Miniscope was running at a lower frame rate than set (18.9 Hz instead of 30 Hz). Moreover, timestamps assigned to the Miniscope frames by Bonsai showed a high jitter, incompatible with the constant intervals between frames. In summary, Syntalos outperforms Bonsai due to its automatic, built-in time synchronization.

## Application case

As an example to validate the capabilities of Syntalos, we performed a learning task with mice in an M-maze[23,24]. In this hippocampus-dependent paradigm, the mouse is supposed to visit each arm of the maze one after another in order to receive a food reward. If one arm is skipped, no reward is dispensed. When moving from one of the outer arms towards the inner arm (inbound trial), correct choice depends mainly on reference memory (rule learning: move inwards after visiting an outer arm). When the animal is moving from the inner arm to the next arm (outbound trial) working memory is crucial, as the animal needs to remember in which of the outer arms it has been in the preceding trial (Fig. 6A, left panel and Supplementary Fig. S6 for the logic diagram and Syntalos configuration). While the animal is performing these trials, various behavioral and physiological parameters can be measured. In our case, mice were equipped with a UCLA Miniscope to record calcium transients in hippocampal CA1 pyramidal cells, and behavior was assessed by video-monitoring (for hardware used, see methods section). For this experiment, pyramidal neurons were intentionally labeled sparsely in order to make it possible to separate them easily and to potentially identify them later using a different microscope, after the animal was perfused and the lens was removed. The sparse labeling was a specific requirement for post-processing steps of this experiment, and does not reflect the norm when performing CA1 recordings with Miniscope[25].

Previous studies using the M-maze have been performed with rats, which easily find the correct strategy[24]. We found that mice, in contrast, learn the paradigm much more reliably if they are initially guided to the correct arms. Therefore, we used custom-made movable gates for conditionally blocking access to the wrong arms. Sweetened condensed milk was given as reward for correct choices. Gates and dispensers were controlled by a Python module. After two days of forced trials, the gates were removed and the mice were allowed to move freely within the M-maze. Syntalos was used to record a video of the behaving animal as well as live images of calcium activity (Miniscope) during the whole trial.

During the experiments Syntalos performed without any issues at synchronizing the camera and the Miniscope as well as at controlling the maze actuators (feeders and gates) and reading the animal's position and information from sensors like light barriers, the Miniscope orientation sensor and illumination sensor. Data analysis was conducted on the Syntalos-generated data in its EDL layout, with calcium activity being extracted as a postprocessing step using the Minian toolbox[26]. Likewise, the animal's behavior and position was tracked offline using DeepLabCut[27]. Learning performance was evaluated using bayesian analysis to estimate learning curves as described by Smith et al.[28]. For the inbound trials, the mouse in our example already learned the task after trial 19 on the first day (certainty >0.95 that the mouse performs better than chance) and on its last day reached more than 90% accuracy. For outbound trials, the animal needed longer time and learned the task on the second day at trial 32, and by the end of the last day 5 performed correctly 80% of the time (Fig. 6A, graphs, right panel). Multiple place cells were found and one that was specific to the middle reward point is depicted in Fig. 6B–E as an example. The firing behavior of this cell remained stable over multiple days. For this to work, high temporal precision and alignment of the top camera images (recording the animal's position) and the calcium recordings by the Miniscope was required and ensured by Syntalos. Using the uniform EDL data structure, an automated pipeline was developed which allowed extracting calcium- and position data from the recorded images in a fully automated way immediately after the experiment was completed. Due to the sensors in the maze itself, the animal's daily performance could also be visualized immediately and tracked in real-time via Syntalos built-in plotting module.

In summary, Syntalos allows to trustily synchronize different devices and thus to reliably and robustly detect complex correlations between behavior and single cell activity. The standardized EDL format additionally facilitates fast and automatic analysis of data.

## Discussion

Synchronization of different devices is a major challenge for complex experiments in modern neurosciences. With Syntalos, we provide a

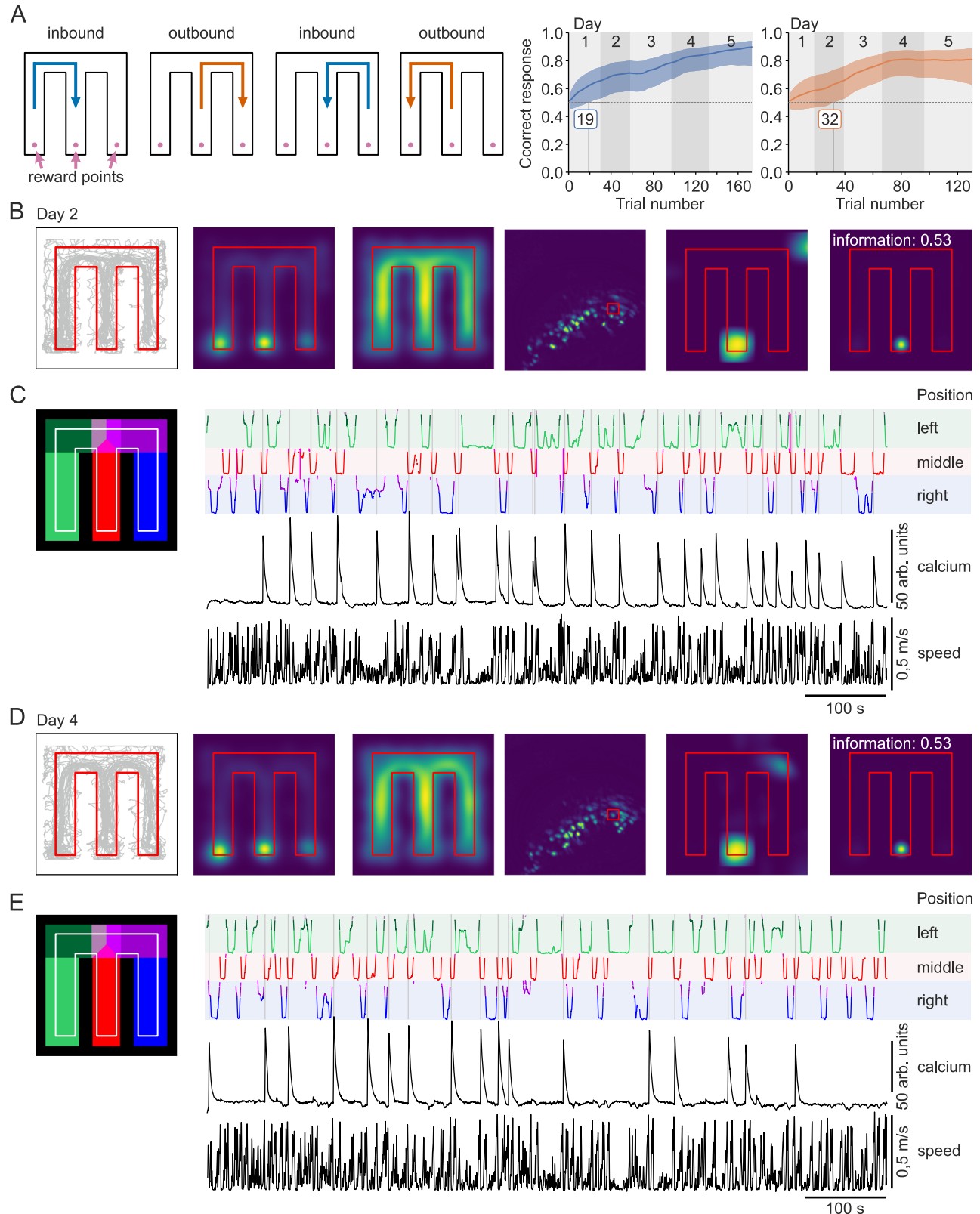

tool for integrating and synchronizing data acquisition pipelines and closed-loop experimental manipulations. Our data show that Syntalos guarantees timestamp synchrony for multiple types of hardware, including devices which do not support external synchronization via a clock pulse. Synchrony is stably maintained over prolonged periods of >24 h. A frequent challenge is that many hardware devices used in neuroscience experiments lack a dedicated input for precise timing by

an external Master clock. Syntalos provides a solution to this problem by continuously measuring and algorithmically correcting timestamp differences between peripheral clocks and the Master clock. Moreover, different devices can be easily incorporated and connected to form complex data acquisition pipelines without extensive programming.

Compared to existing tools for data acquisition, such as ANY-Maze (Dublin, Ireland), Noldus EthoVision XT (Wageningen,

**Fig. 6 | Example application of a behavioral experiment with simultaneous recording of calcium signals with an UCLA Miniscope. A** M-maze rules for successful trials and learning curves for inbound (left) and outbound (right) trials over several days. **B** Experiment day 2, example analysis. First panel depicts the trajectory of the animal. Second panel: probability to find the animal at a certain location. Note that the animal spends more time at the reward points than on the different tracks/arms. Third panel: animal speed; in line with the previous panel. The speed in highest in the straight arms and lowest at the reward points Fourth panel: maximum intensity projection of the calcium imaging. Cells that were active during the experiment are visible in the image. Fifth panel: Conditional firing probability of the cell outlined in the previous panel. The cell is a place cell that is specific to the reward point at the middle arm. Last panel: mutual Shannon information of the location and the activity of the cell. **C** Visualization of the trajectory of the animal and the calcium signal. Left panel: color coding of the individual sections of the M-maze. Linear coordinates are additionally assigned to each arm. Right panel: linear coordinates of the animals position (three upper panels), the 'raw' calcium signal (quantified by Minian after motion correction and ROI assignment) of the cell highlighted in **B**, and the speed of the animal (two lower panels). Note that the cell is activated when the animal enters the reward location in the middle arm. **D** and **E** similar to **B** and **C** but on experimental day 4. Note that the cell from day 2 is also active on day 4 and has the same firing field and information content.

Netherlands) or Bonsai (https://bonsai-rx.org/), Syntalos is the only software that uses a statistical method for timestamp synchronization, while the other solutions rely on TTL pulses and manual alignment or dedicated hardware. This has the important advantage that some widely used devices which do not have a TTL output or input can be synchronized. For example, the UCLA Miniscope has only a TTL output but no input. Therefore, depending on the combination of devices, synchronization cannot be accomplished by TTL pulses and there will be uncertainty as to whether timestamps are correct. This, of course, additionally allows our proposed method to be used by other scientific disciplines where the exact synchronization of different data sources plays a crucial role and hardware solutions are not possible.

From all recording programs, Bonsai comes closest to Syntalos, as it supports multiple open-source hardware devices, is used for closed-loop interactions, and is open-source itself. We therefore performed a direct comparison between both programs. Syntalos outperformed Bonsai in various aspects, mostly due to its independence from external timing cues. This was, in our recordings, most apparent for the Intan DAQ board and for the UCLA Miniscope (see Supplementary Fig. S5). We clearly acknowledge the strengths of Bonsai, e.g. its graphical programming capability and its good applicability in behavioral experiments. Syntalos, however, meets the needs for precise synchronization of a very broad range of data sources and devices, independent from external cues and during long recording periods. Several further features of Syntalos make it a particularly versatile and user-friendly tool for multi-modal experiments, as shown in a systematic comparison with existing systems (Supplementary Table S4).

Syntalos relies heavily on multithreading. This design facilitates quick reactions to new input events and prevents tasks being blocked by each other over CPU resource demands. On the other hand, this strategy can be computationally inefficient: CPU time may be used by multiple threads even when these are idling, and CPU caches must be updated whenever the operating system switches between threads. These potential disadvantages were, however, no limiting factors for Syntalos' performance. To test for potential shortcomings, we performed test-runs on old computing hardware with relatively slow CPU- and memory-performance and did not observe problems with data acquisition and synchronization capacities (see 'System Requirements' section in supplemental information). A further requirement is that threads must synchronize whenever they access and overwrite the same memory. We therefore used a lock-free queue for message passing. This minimizes the amount of cycles wasted when a large amount of data is transferred between Syntalos module threads. Efficiency was further optimized by reducing memory allocations during data transfer, thereby reducing additional latencies.

Alternative solutions to our multithreading approach are asynchronous parallelization paradigms, e.g. reactive programming or event-based programming[29,30]. This design optimizes CPU usage by reducing idle time and is a lot more efficient. It bears the risk, however, that tasks block each other for prolonged periods which would compromise precise timestamping or synchronization. Thus, such approaches are ideal for processing multiple similar or identical tasks, especially when CPU capacity is limiting. Typical experimental setups

in neurosciences do not meet these criteria and, hence, allow to make use of a more threading-focused design.

Behavioral or optogenetic experiments make increasing use of real-time interventions. Behaviorally relevant time scales are usually in the range of tens of milliseconds (with rare exceptions like the jump takeoff of *Drosophila melanogaster* which happens in 5 ms), setting relatively low demands on temporal precision[31–33]. Some closed-loop interventions, however, require millisecond precision and highly constant latencies between cue and response[34]. Syntalos, by its design, is not ideally suited for such hard real-time constraints. Memory allocations as well as inter-thread and inter-process communication cause small but not precisely predictable delays which can add up to significant latencies. In our experiments, such latencies were in the range of 2–6 ms for standard Python code, and ~150 μs when using Syntalos with MicroPython (Fig. 5D). The standard Python module in Syntalos may not be sufficiently precise for interfering with very fast processes like sharp wave-ripple oscillations[35–37], while MicroPython can be limiting for such applications as well. In such cases, we suggest using dedicated software on a separate computer or a FPGA device, which is then controlled by the DAQ system (for an example, see Fig. S4).

Syntalos, unlike any other multi-modal DAQ solution, runs on Linux, such that all parts of the data acquisition pipeline are open-source and freely modifiable. During development, the simple drag & drop GUI was refined in several iterations with feedback from inexperienced or new users of different laboratories, resulting in an easy-to-use interface for individualized experimental designs. During these interactions, special emphasis was put on stability during ongoing experiments, automated resource monitoring and efficient warnings to prevent data loss in failure scenarios (e.g. lack of hard disk space for data storage). For test and demonstration purposes, Syntalos can also be run on the Windows subsystem for Linux (WSL2), such that users can explore the program without changing their operation system to Linux. However, we do not recommend this version for real experiments.

Overall, Syntalos provides a versatile and easy-to-use tool for the scientific community dealing with complex experimental settings, especially those combining behavioral studies with physiological measurements. Syntalos ensures easy data acquisition and produces standardized, shareable raw data. Instead of replacing existing DAQ solutions, it can integrate and synchronize multiple existing acquisition tools, facilitating data acquisition for (neuro)scientists in a broad range of applications (see Figs. 1, 6 and Supplementary Fig. S7 for application examples), and is already in active use[38]. We envision that our Syntalos framework will be broadly applicable to enable and scale up precise correlations between behavior and the underlying single cell and neural network activity to find novel mechanisms how the brain computes and processes sensory stimuli by its internal representations and how such representations can lead to targeted behavior.

## Methods
### Ethical statement
Animal treatment and all experimental procedures were approved by the state government of Baden-Württemberg and performed in

accordance with regulations and guidelines of Heidelberg University and the Federation of European Laboratory Animal Science Associations (FELASA) under the supervision of local ethics committees (permission by the state government of Baden-Württemberg, no. 35-9185.81/G-62/19, 35-9185.81/G-216/19, and 35-9185.81/G-78/23).

The design and use of Syntalos are described in the Results section. Test runs of the software suite were mostly performed with technical equipment, not involving any biological material or animals. Realistic 'use case' experiments have been performed with mice (see Results and Figs. 1, 6, Supplementary Fig. S7, and ref. 38).

### Software bill of materials

Syntalos is written in modern C++20 using the Qt framework. It runs on the Linux operating system, requiring at least kernel 4.20 (all tests were run on systems with Linux 5.10 or higher). The software components shown in Supplementary Table S2 are directly used to build and run Syntalos. Some modules require dedicated software components or embed additional libraries (see Supplementary Table S3).

### Electrophysiology behavior experiment

For the data presented in Fig. 1, we used six adult male C57BL/6NRj mice (Janvier Labs, Le Genest-Saint-Isle, France), aged 8–10 weeks at the onset of training. The animals were individually housed in a ventilated Scantainer (Scantainer Classic, SCANBUR A/S, Karlslunde, Denmark) maintained under controlled conditions: a 12-hour inverted light/dark cycle (lights off at 7:00 a.m. and on at 7:00 p.m.), temperature range of 22–25 °C, and humidity levels between 40 and 60%. During behavioral training (conducted during the light phase), a food restriction protocol was followed to maintain each animal's body weight at 85–95% of its baseline weight, with ad libitum access to water.

In preparation for surgical procedures, buprenorphine hydrochloride (Bupresol vet. Multidose 0.3 mg/ml, 10 ml, CP-Pharma Handelsgesellschaft mbH, Burgdorf, Germany) was administered subcutaneously at a dose of 0.1 mg/kg of body weight, 30 min before general anesthesia. Anesthesia was induced and maintained with isoflurane (1–2% in oxygen, Isofluran Baxter, Baxter Deutschland GmbH, Germany), and depth was adjusted according to the eyelid and pedal reflexes. Throughout surgery, body temperature was kept between 36–38 °C using heating pads. To protect the eyes, a corneal application of eye ointment (Bepanthen®, Bayer, Germany) was applied, and subcutaneous saline (30 ml/kg body weight, Fresenius Kabi Deutschland GmbH, Bad Homburg, Germany) was administered to maintain hydration. Lidocaine (Xylocaine® 1%, Aspen Pharma Trading Limited, Ireland) was used to anesthetize the scalp locally before surgery. The animal was then positioned in a stereotactic apparatus (David Kopf Instruments, Tujunga, CA, USA) designed to minimize trauma. Small holes were drilled into the skull over the designated target areas (coordinates available in Supplementary Table S6, tetrodes made from 12.5 μm diameter tungsten wire, California Fine Wire, CA, USA) using a dental drill (78001 Microdrill, RWD Life Science, TX, USA). The electrode interface board (EIB) was lowered to the desired depth and affixed to the skull using cyanoacrylate adhesive (Super-bond, Sun Medical, Japan) and dental cement (Paladur®, Kulzer, Germany). After the procedure, animals were placed on a heated surface in a pre-warmed enclosure before being returned to their ventilated housing. Behavioral experiments started only after a full recovery period following EIB implantation.

Whisker touch responsiveness was defined by a significant change in firing rates upon aperture touch within a 200 ms response window following aperture touch (to either of the two aperture states), analyzed using a two-sided Wilcoxon signed rank test.

Aperture state decoding from neural spike timing data was conducted with the Neural Decoding Toolbox[39]. Classifier training was performed by splitting the dataset into a training set (90% of labels) and a test set (10% of labels). Each set included spike traces from individual neurons for a given trial along with the corresponding aperture labels. Training of a support vector machine (SVM) classifier was carried out using the LIBSVM software[40]. The classifier underwent 10-fold cross-validation, ensuring independent training and testing across different data partitions. To evaluate decoding accuracy variability, each selection of units was bootstrapped 20 times. Decoding accuracies in the box plots were calculated over a time window ranging from trigger onset to 400 ms after trigger onset. Data are represented using box-and-whisker plots, where the central line within each box denotes the median value. The upper and lower boundaries of each box correspond to the upper (75th percentile) and lower (25th percentile) quartiles, respectively. Whiskers extend to the furthest data points within 1.5 times the interquartile range (IQR) from the edges of the box, capturing the maximum and minimum values that are not considered statistical outliers.

Surgery procedure for data shown in Supplementary Fig. S7 was similar but instead of tetrodes, single electrodes made from 50 μm diameter tungsten wire (California Fine Wire, CA, USA) were used. Coordinates are available in Supplementary Table S7.

### Miniscope behavior experiment

Figure 6 shows an example experiment of a male C57/Bl6N mice (WT, Janvier Labs, Le Genest-Saint-Isle, France) implanted with a UCLA Miniscope v4. The surgery was performed at age P42 in two steps. First, a GRIN (gradient refractive index) leans (GRINTECH, NEM-100-20-20-520-S-0.5p) was coated with an emulsion of Fibroin[41], 1:8000 diluted Cre-virus under the CamKII promotor (Addgene, AAV1.CaMKII 0.4.Cre.SV40) and GCaMP-7f-containing virus (Addgene, AAV1-syn-FLEX-JGCaMP7f-WPRE) and implanted above the right hippocampus after the cortex above the implantation site was removed. After three weeks of expression, a second surgery was performed to implant the base plate of the UCLA Miniscope at a position optimized to the expression of GCaMP. After the second surgery, the animal was allowed to recover for another week before starting the behavioral experiments. During the time after the first surgery and the beginning of the behavioral experiments, the animal was gradually accustomed to the experimenter and the experimental room.

The behavioral experiment was performed on an M-maze (Fig. 6A). For getting reward (15 μl of condensed milk mixed 1:1 with water for final concentration of 4 % fat, 10 % fat-free dry milk, 27% sugar) at the endpoints of the individual arms, the animal has to alternated between the arms i.e. in the following fashion: left → middle → right → middle → left and so forth. The reward was given with custom-made automatic dispensers, which achieve micro-liter accuracy (https://github.com/bothlab/maze-hardware/blob/main/README.md). The behavioral experiment consisted of a session of 15 min over seven days. During the 15 min, the animal could freely explore the maze. However, during the first two days, gates at the connection of the three arms forced the animal to the correct arm. Thus, animals were familiarized with getting reward at the endpoints of the arms and the required rule. Those two days are omitted in Fig. 6. After these two days, the animal could freely choose between the arms, but was only rewarded when the sequence was correct. The performance of those five days is depicted in Fig. 6A.

Calcium signals were analyzed using Minian[26]. The position of the animal was determined using DeepLabCut[27].

### Study design and consideration of sex

Building on a body of work conducted in male subjects, this study was designed with a focus on male subjects only, in order to ensure comparability and reproducibility with prior research and established protocols. Furthermore, this study focuses on validating the functionality and reliability of the newly developed software. The software is designed to be broadly applicable, and future studies will explore its use across both sexes.

**Reporting summary**

Further information on research design is available in the Nature Portfolio Reporting Summary linked to this article.

## Data availability

The datasets used to validate Syntalos in this study are available on Zenodo at https://zenodo.org/records/13862969. Additional datasets and code to create the figures for this study are available at https://zenodo.org/records/14179869.

## Code availability

Code created for and referenced in this study is available freely under Open Source licenses: Syntalos: https://github.com/syntalos/syntalos[42]. PoMiDAQ (Miniscope support): https://github.com/bothlab/pomidaq[43]. Edlio (Python data reader for EDL): https://github.com/syntalos/edlio[44]. Timing-validation (time synchronization analysis code used in this study): https://github.com/syntalos/timing-validation[45]. LaBrStim (GALDUR online stimulation code): https://github.com/bothlab/labrstim[46]. Maze Hardware Designs: https://github.com/bothlab/maze-hardware[47]. Syntalos is also made available in the Flathub Linux software store, for easy installation on any Linux distribution without the need to compile any code. We also provide packages for Debian and a binary package repository for Ubuntu. You can find out more at: https://flathub.org/apps/org.syntalos.syntalos and https://syntalos.org/docs/setup/install/.

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

## Acknowledgements

This work was supported by the German Research Foundation (grant SFB1134, project A03 and DFG BO 3512/2-1 to M.B. as well as DFG Grant GR3757/4-1 to A.G.). The authors acknowledge support by the state of Baden-Württemberg through bwHPC and the German Research Foundation (DFG) through grant INST 35/1597-1 FUGG. We thank Nadin Saluti and Melina Castelanelli from the Alexander Groh group, the Thomas Kuner group, and Avi Adlakha for testing various versions of the Syntalos software and providing valuable feedback. We also thank the developers of all open-source projects we depend on for providing insights, merging our patches or addressing bug reports, as well as for providing the building blocks to create Syntalos in the first place. Syntalos itself as well as Figs. 1, 2, 3, 5, Supplementary Figs. S3, S4, S5 and S7 contain graphics based on Breeze icons copyright KDE and licensed under the GNU LGPL version 3 or later https://develop.kde.org/frameworks/breeze-icons/. Source files for the icons are available in Syntalos' code repository. "Python" and the Python logos are trademarks or registered trademarks of the Python Software Foundation. The Intan and MicroPython logos belong to the respective organizations and are used with permission. The Bonsai logo is published by the Bonsai Foundation CIC and distributed under the CC BY-SA 4.0 license. The Arduino logo is used for explanatory purposes only. The Aravis icon was created by Emmanuel Pacaud and published under the CC BY-SA 4.0 license. The Linux mascot (the Tux penguin) can freely be distributed. We use a redrawing of Tux from the Breeze icon set with permission. The Miniscope logo belongs to the Miniscope Project, published under the GPL-3.0 license. Figures 1, 3, 4, 5, 6, Supplementary Figs. S4 and S5 are made using Matplotlib[48]. For body part tracking we used DeepLabCut (version 2.3.0), the DeepLabCut logo used in Supplementary Fig. S3 belongs to the DeepLabCut project, licensed under the LGPL-3.0 license[27,49].

## Author contributions

M.K. designed software, M.K. and L.E. designed hardware, M.K., A.G. and M.B. conceived of and designed experiments, M.K., F.H., A.L.A.D., and J.S. performed experiments, M.K., M.B., and F.H. analyzed the data, M.K., A.D., and M.B. wrote the original manuscript. All authors revised and edited the manuscript.

## Funding

## Competing interests

The authors declare no competing interests.
