## [Transparent Peer Review file · Nature Communications]

Syntalos: A software for precise synchronization of simultaneous multi-modal data acquisition and closed-loop interventions

Corresponding Author: Dr Martin Both

Version 0:

Reviewer comments:

Reviewer #1

(Remarks to the Author)

The authors have identified a standard problem facing in vivo neurophysiologist, namely synchronising multiple recording techniques within a single experiment, for example behavioural tracking, electrophysiology/miniscope and optogenetics. They provide a new platform to achieve this. In describing the new platform, the authors provide a robust validation of how "Syntalos" synchronises multiple input streams, as well as a characterisation of timing drift of recording systems in common use. However, to date, multiple labs have solved this "synchronisation" problem, leaving a question over whether this system represents a significant advance. Moreover, the closed loop functionality of the system is limited by a ~30ms (or greater) latency between a detected stimulus and initiating an intervention.

Main concerns

1) Synchronisation of the recording system and behavioural tracking or events within a behavioural protocol is a problem that has been solved by multiple labs (e.g.(1)). In addition, commercial systems are available that provide similar functionality, for example, Plexon systems (<https://plexon.com/plexon-systems/get-started-with-omniplex/>). While an open source and carefully validated platform offers a free alternative, The platform may offer a limited advance on previously implemented systems.

2) Closed loop systems has been have been implemented by many labs. For example, Gridchyn et al. (1) implemented a closed loop system that used electrophysiological signals to trigger optogenetics. Here, the reaction time of the closed-loop system (time from event detection to a detectable change in the local field potential by a light pulse) was ~1.04 ms. This delay included data acquisition, spike detection, feature extraction, and online decoding of firing patterns. The authors report a apparently variable delay of ~30ms, which is substantially above that implemented by Gridchyn and colleagues. Such a delay would be problematic for projects aimed at driving interventions timed to fast events such as gamma oscillations or Sharp Wave Ripples.

Minor concerns:

1) The authors have implemented the system in Linux. They report that this approach allows use of single PC to safely handle multiple tasks with high overheads. However, Linux may be off putting to some researchers unfamiliar with this operating system, which may form a core of the target user group.

2) Multiple methods, or "synchroniser constructs", were used for aligning clocks across modules. If each method is as robust as the other, why not choose a single method to synchronise the data, e.g. "write the offset information to a special tsync binary file"? The authors approach seems unnecessarily complex, which could increase the likely hood of errors/bugs. Gridchyn I, Schoenenberger P, O'Neill J, Csicsvari J. Assembly-Specific Disruption of Hippocampal Replay Leads to Selective Memory Deficit. Neuron. 2020 Apr 22;106(2):291-300.e6.

(Remarks on code availability)

Reviewer #2

(Remarks to the Author)

In this manuscript the authors describe a Linux based software for synchronization of data acquisition systems, called Syntalos. It does this by checking and aligning the acquisition timestamps to a master clock from the Linux computer. It also allows to control actuators within 30-50ms.

The technical details of the system as well as the benchmarking tests to assess synchronization stability are very well described. The authors present one experiment from one mouse in a maze recording calcium imaging and video as application case. Another example of electrophysiological recordings is used to illustrate the risk of misalignment, but it does not seem to have utilized the system.

The authors indicate that Syntalos could be used for “multiple experimental approaches” for “real time closed experiments”. In the abstract, it is mentioned that “an arbitrary number of sources, including multi-channel electrophysiological recordings and different live imaging devices, as well as closed-loop, real-time interventions with different actuators.” However, the electrophysiological equipment is only used during benchmarking and not in a real experiment (unless I misunderstood, and it was used in the Fig. 1 Ephys experiment) and the closed loop described has a ~30ms and not real time response.

The application case presented is well described, but it is also very specific to the problem in hand (e.g., in house designed actuators). For the system to be widely adopted by the neuroscience community, additional examples and a broader range of tested sources and actuators is important.

More specifically:

- To ensure that the title reflects the nature of the system I would suggest adding “synchronization”. For example: “Syntalos: A software for precise synchronization of simultaneous multi-modal....”

- Similarly, in the abstract, it should be made clearer that Syntalos is a software to synchronize data. Please also indicate that the experiments were performed with mice. This might not be obvious outside the neuroscience community.

- Closed-loop. As the authors indicate, ~30ms latency is too high for many closed-loop interventions. It is also mentioned that lower latency could be achieved with C++ modules removing the overhead of using Python modules. What latency could be achieved? This is not established. Could you provide benchmarking information for such a closed-loop system? (or even better, an experimental example). One idea would be to repeat the example shown in Fig.5.D with a C++ implementation to compare the two approaches.

- It is my understanding that the initial example on mice touching apertures of varying widths was not acquired using Syntalos. How was it acquired and synchronized? Fig 1.C and 1.D seem to suggest that the paradigm was implemented on Syntalos. If it was it would be an interesting example and comparison to the original one. If it was not, this or another Ephys experiment would be interesting.

- Additional application cases. The abstract mentions that “preliminary experiments with different research questions...” But only 1 application case is shown. Please add more. In line with the above, the authors also mention that they obtained feedback from different laboratories, which allows me to assume that they could provide a larger number of application cases. Presenting more than 1 example and a comparison with other systems would demonstrate the usability and increase the adoption of Syntalos.

- Comparison to existing systems

- o In the introduction other software (Bonsai RX, ANY-Maze or Noldus EthoVision XT) is listed and a list of key requirements indicated. It would help to have references for each software and to indicate which of the key requirements each one covers and which ones they do not. This will help understand what Syntalos brings that is different to the others.

- o In addition, in the supplementary data there is a comparison to some of them which could be moved to the discussion.

- o Furthermore, there is no comparison to other systems commonly used in neuroscience such as OpenEphys (e.g., with the video frame grabber plugin or Bonsai). Please add some additional comparisons (e.g. to Tarcsay et al., eNeuro 2022 and Buccino et al., J Neural Eng 2018).

- o How different it is from a Simulink Real-Time or Labview implementation? These in principle would also allow to impose a fix timestamp and also provide a block diagram implementation.

- o Ideally a direct comparison with one or some of these systems (most interestingly with Bonsai) would help understand the advantage of using Syntalos.

System capabilities and characteristics.

- Other devices. Could you indicate more clearly what the process is for getting timing information from devices without a clock signal?

- Could it record from multi subjects? How would this be achieved? (e.g. Kimchi et al., eNeuro 2020).

- What would happen if a device were disconnected halfway through the experiment? Would it keep running? Would the other modules stay connected?

- Line 344 says that “performance could also be visualized immediately and tracked in real-time”. How?

- EDL format. Please compare to standard formats. Why is this data structure selected instead of already implemented formats in neuroscience?

Additional minor suggestions:

- A couple of videos (could be screen recordings) demonstrating how to use Syntalos would be helpful.
- There are no references in Figure 4 (I understand that they are the same as in Fig 3 – but please add them).
- Fig 6. A diagram of Syntalos in this experiment would help understand how to use it.
- Lines 100-107 seem to be part of the general architecture.

(Remarks on code availability)

I could not install the software on my available Linux machine (the required operating system), but I looked at the code and it seems well structured and documented.

Reviewer #3

(Remarks to the Author)

Modern neuroscience requires progressively more complex experimental setups that incorporate multiple modalities for recording or manipulating neurons while capturing and controlling animal behaviour with sub-second time resolution. Such complex and custom-made setups would benefit greatly from using a software tool that combines multimodal data acquisition and control, ensuring a high degree of synchrony between data sources. Klumpp et al. present a Linux-based open-source software solution for this, called Syntalos, which offers an easy-to-use interface that allows highly precise and synchronous multimodal recordings and closed-loop device interactions. The software presented serves an important niche as most competing software relies on the restrictive operating system environment of Microsoft Windows, while Syntalos is Linux-based. Also, as our lab and those of colleagues' have experienced time and again, multimodal data synchrony is key to more and more experiments but often difficult to achieve when operating custom-made setups consisting of diverse components. This is an exciting manuscript presenting a software tool that will enable a growing number of labs to record and control such multimodal experiments, while ensuring synchrony between data modalities. The figures and text provided by the authors largely allows a non-expert audience to understand the rationale behind Syntalos, how it works, and what it achieves. I believe this manuscript is, overall, well suited for publication in Nature Communications, but I have a few suggestions that would increase the impact of both the manuscript and the accompanying software:

Major:

1. Syntalos is, of course, not the only software solution for multimodal data acquisition and closed-loop experimental control. The authors rightly acknowledge the widely used free tool Bonsai RX and commercial solutions ANY-Maze/EthoVision. I appreciate the gap Syntalos is filling here in both providing a Linux version (all other software solutions work on Windows only) and investing considerable effort in establishing a user-friendly interface with a high degree of synchrony between data sources. Both specificity and added value should be made clearer though. I suggest incorporating "Syntalos: A Linux-based software..." into the title so that potential users can easily find it and die-hard Windows users will know this won't work on their system right away. Furthermore, as with any piece of software that has comparable competitors, I strongly suggest the authors quantify their comparisons instead of just drawing hypothetical comparisons. Specifically, I suggest the authors compare multimodal synchrony across the 4 different software tools. In addition, a table listing the features Syntalos provides and the OS it operates on besides those of the other competing pieces of software would be beneficial.
2. I commend the authors for establishing a systematic experiment directory layout (EDL), but it is not clear to me what the benefits of EDL are compared to the more and more widely used Neurodata Without Borders (NWB) format that aims to unite neurophysiological and behavioural data across labs. I believe Syntalos and the wider open data community would benefit greatly if Syntalos (also) offered to write and read data in the NWB format, unless I'm missing something here (in which case, this should also be pointed out in the text). Given that the EDL format has been developed in-house and has no associated publications yet as far as I'm aware, some introduction to EDL and why it is being used here should also be provided.
3. Syntalos has a potentially very broad appeal to many labs performing *in vivo* neurophysiology experiments based on its various modules, yet the authors show only data of a (de-)synchronised electrophysiology/behaviour experiment (Figure 1A-C) and a miniscope/camera-based locomotion experiment (Figure 6). Given the prevalence of (1) two-photon / head-fixed camera / analogue behaviour recordings and (2) optogenetic stimulation / freely moving behaviour, I suggest the authors include at least one further set of one or both modalities into their data, showing the respective Syntalos configuration required, and how the acquired data looks, and how desynchronisation can be ruled out.

Minor:

4. Although not strictly part of the manuscript, I believe the software would find wider acceptance if the authors incorporated further improvements into their documentation (<https://syntalos.readthedocs.io/>), such as (1) a simplified installation (I had trouble with Flatpak on Ubuntu 22.04 and the package-based installation notes say "After the PPA is registered..." without making mention of the PPA address), (2) releasing video tutorials instead of text/graphic only, and (3) completing some important documentation content, e.g. Introduction "Design Goals: Coming soon! Architecture Overview: Coming soon!".

5. I understand the miniscope data in Figure 6 is mostly for demonstration purposes only, but having acquired miniscope data in dCA1 for several years, I am not convinced by the quality of data presented. This is hard to judge from the limited amount of actual calcium data presented (some raw calcium traces with position/velocity data should be presented), but the fourth panels from the left in 6B and 6D suggest a strongly oblique field of view on dCA1 that would allow the capture of only a comparatively small number of neurons. As a point of reference, I suggest the authors inspect data and protocols from Yaniv Ziv's recent work (Geva et al., 2023, Neuron, <https://doi.org/10.1016/j.neuron.2023.05.005>, e.g., Figure 2C).
6. It is not entirely clear to me how Syntalos performs mouse tracking as for Figure 6, as depicted in Figure 1D to the left. Is this based on a DeepLabCut module? Does it require a pretrained DLC model? Or is it based on background contrasting? Does it allow online device control? The authors should make clear how this works to boost readers' understanding how it could be useful for them.
7. I have come across some typographic mistakes: (line 35 "Jensen", Fig 2 "Syntlos", Fig 5 "guarantied").
8. The full name for "IPC" (mentioned in Figure 2) should be stated.
9. Fig 2B: It is unclear what "Engine" and "Module" explicitly refer to at top / bottom. Could the authors clarify how this figure is set up?
10. Could the authors provide a table/list which types of cameras are currently supported? Our lab and those of colleagues largely rely on Basler cameras for example. Are these supported?
11. Could you please improve either the way Figure 4E right panel displays the data or how it is described? As it stands, I could not grasp what is depicted, unfortunately.

(Remarks on code availability)

Reviewer #4

(Remarks to the Author)

I co-reviewed this manuscript with one of the reviewers who provided the listed reports as part of the Nature Communications initiative to facilitate training in peer review and appropriate recognition for co-reviewers.

(Remarks on code availability)

I did not have access to a Linux machine to run the software but was able to confirm access to the github database which had code available for download and a corresponding flathub page which aids in distribution of Linux software.

Version 1:

Reviewer comments:

Reviewer #1

(Remarks to the Author)

The authors have provided a new and exciting platform for acquiring multiple complex data streams from in vivo experiments. My concerns were centered on the novelty of the proposed system, whether the system could provide optimal real time closed loop functionality and whether the system could provide sufficient user friendliness to be useful to the wider community. The authors have largely addressed these concerns in their response to the reviewers, as well as upgrades to the software.

In particular, the authors have significantly improved the real time aspect of the software, and in doing so made cutting edge methodology available to the wider community. Some effort has also been made to improve how users can install the system, which increases the reach of the system to a wider audience. I completely agree that implementation on Linux offers a big advantage to many. However, the impact of the software depends on whether some of those users would instead opt for in house, "home brew" solutions, tailored to their specific experimental setup. This speaks to the critical question of novelty. The authors present a strong case for the advantages of their system over expensive and potentially restrictive commercial packages, while also demonstrating how their software provides significantly greater functionality than individual free software. Given this, I believe that Syntalos will make an important contribution to in vivo research.

(Remarks on code availability)

Reviewer #2

(Remarks to the Author)

I would like to commend the authors for the amount of work they performed to address the reviewers' concerns. In particular, the changes to address latency and the comparison to other systems are much appreciated. My concerns have been addressed.

(Remarks on code availability)

Reviewer #3

(Remarks to the Author)

The authors have now addressed all of my concerns, and I have no more concerns regarding the publication of this manuscript. I congratulate the authors on a very useful piece of software that is now well-described and placed into context.

(Remarks on code availability)

Reviewer #4

(Remarks to the Author)

(Remarks on code availability)

I did not have access to a Linux machine to run the software but was able to confirm access to the github database which had code available for download and a corresponding flathub page which aids in distribution of Linux software. Additionally, a thorough documentation page is available including detailed installation instructions.

Dear reviewers,

Thank you for reviewing our manuscript ‘Syntalos: A software for precise synchronization of simultaneous multi-modal data acquisition and closed-loop interventions’ (title has been revised as per suggestion of reviewer 2). Following the reviewers’ suggestions, we have substantially changed the Syntalos software suite as well as its description and data analysis. Changes include technical advances (e.g., implementation of strongly reduced closed-loop reaction times, adding new drivers for relevant camera systems, and an option to continue recording after failure of single devices), more extensive comparisons with existing systems, increased description of applications, an improved documentation by text and video in public repositories, more application examples, and multiple changes in text and figures.

We feel that Syntalos has been strongly improved through these measures, and we thank the reviewers for their thoughtful and constructive comments, as well as for their positive and encouraging overall assessments.

In the following, we address their concerns point by point. Issues are numbered in the order of appearance in the reviews.

Reviewer 1:

The authors have identified a standard problem facing in vivo neurophysiologist, namely synchronising multiple recording techniques within a single experiment, for example behavioural tracking, electrophysiology/miniscope and optogenetics. They provide a new platform to achieve this. In describing the new platform, the authors provide a robust validation of how “Syntalos” synchronises multiple input streams, as well as a characterisation of timing drift of recording systems in common use. However, to date, multiple labs have solved this “synchronisation” problem, leaving a question over whether this system represents a significant advance. Moreover, the closed loop functionality of the system is limited by a ~30ms (or greater) latency between a detected stimulus and initiating an intervention.

We thank the reviewer for the insightful comments and suggestions and hope to have addressed his/her concerns in the specific points below.

Main concerns:

1) Synchronisation of the recording system and behavioural tracking or events within a behavioural protocol is a problem that has been solved by multiple labs (e.g.(1)). In addition, commercial systems are available that provide similar functionality, for example, Plexon systems (<https://plexon.com/plexon-systems/get-started-with-omniplex/>). While an open source and carefully validated platform offers a free alternative, The platform may offer a limited advance on previously implemented systems.

We agree that various solutions to this problem are available. However, we are convinced (and hear from our colleagues) that there still is great demand for a flexible, multi-purpose solution for multimodal data acquisition that works without post-processing steps. This is even more

pressing when experiments are complex and require a wide variety of electrophysiological, optical, and other devices from different manufacturers. Precise synchronization of these heterogeneous data streams is still a major problem, especially for long observation periods and when no external synchronization signals can be used. In addition, an ideal system should have versatile closed-loop intervention tools. To the best of our knowledge, there is no such system with similar flexibility of integrating different devices, data types and experimental protocols, especially not with the user-friendly installation and high synchronization precision of Syntalos.

Moreover, proprietary software packages are often expensive, do not permit modification of their built-in algorithms and support only a limited range of external devices (for example, Plexon's OmniPlex requires not only buying the software package but also using additional hardware from Plexon). Plexon has been specifically designed for electrophysiology and online spike sorting, while Syntalos is far more generic. Bonsai is a powerful open-source solution for behavioral recordings. However, it does not provide built-in automatic and precise time synchronization like Syntalos. Also, it is a graphical programming language, while Syntalos offers a far simpler, user-friendly approach to design in-vivo experiments without advanced programming skills.

The need for a new system is now better explained in the Introduction on page 3, and the comparison with other systems (esp. Bonsai, including newly generated test data) is described in much more detail on pages 10, 12 and Figure S5.

2) Closed loop systems have been implemented by many labs. For example, Gridchyn et al. (1) implemented a closed loop system that used electrophysiological signals to trigger optogenetics. Here, the reaction time of the closed-loop system (time from event detection to a detectable change in the local field potential by a light pulse) was ~1.04 ms. This delay included data acquisition, spike detection, feature extraction, and online decoding of firing patterns. The authors report a apparently variable delay of ~30ms, which is substantially above that implemented by Gridchyn and colleagues. Such a delay would be problematic for projects aimed at driving interventions timed to fast events such as gamma oscillations or Sharp Wave Ripples. (Gridchyn I, Schoenenberger P, O'Neill J, Csicsvari J. Assembly-Specific Disruption of Hippocampal Replay Leads to Selective Memory Deficit. Neuron. 2020 Apr 22;106(2):291-300.e6.)

We thank the reviewer for this important hint and agree entirely with the need to shorten the reaction time. We have now re-written the Syntalos' message-passing engine for modules that run as external processes (like the Python module in our example). The new engine is built upon Iceoryx (<https://iceoryx.io/latest/>), a system also used in cars and robotic automation to glue individual processes together. Furthermore, multiple other optimizations have been applied to improve latencies. The roundtrip latency for a Python script running in Syntalos and communicating with an Arduino Uno via the Firmata protocol is now down to 2-6 ms (shown in Fig. 5). The remaining latency is primarily due to the Arduino itself and its connection to the computer (the USB connection confers significant delays when the computer is busy by recording data from multiple devices; see Fig. 3). We note that the Arduino Uno can not communicate via USB and read data at the same time, inducing significant latency in closed-loop applications. Using just an Arduino Uno on its own, others have measured an average roundtrip

latency of ~3.16 ms as well (see <https://github.com/seeraven/arduinoLatency>). For devices communicating with the computer via USB, using a real-time Linux kernel would improve the USB-induced latency and time-variance as well. Specialized external hardware can have significantly lower latencies, as we have demonstrated with the Raspberry Pi Pico which Syntalos controls with MicroPython. For this device, we now measured latencies of 50-150 μ s (the new data are shown in Fig. 5).

Information on the hardware used by Gridchyn et al. is unfortunately a bit light, but given their code at https://github.com/igrichyn/lfp_online, they appear to have used a dedicated computer with a realtime CentOS Linux. Their solution, unlike Syntalos, does one specialized task very well, while Syntalos is trading off the high speed of specialized code with flexibility and usability, allowing anyone to easily set up new experiments with different hardware.

If an extremely low latency is required, Syntalos can control dedicated hardware to perform this task (as shown in newly added Fig. S4 using as Raspberry Pi 4 and a custom interface board). Here, the dedicated hardware performs data acquisition, digital filtering, and spike detection, and the latency mainly depends on the signal length that has to be evaluated by the digital filter (31 samples, i.e. 1 ms @ 30 kSamples/s for a band-pass filter with 0.7-5 kHz). For lower frequencies or frequency contents (e.g. ripple oscillations, ~200 Hz, i.e. 5 ms per cycle, several cycles required to detect a ripple complex), the latency will be correspondingly longer.

Minor concerns:

1) The authors have implemented the system in Linux. They report that this approach allows use of single PC to safely handle multiple tasks with high overheads. However, Linux may be off putting to some researchers unfamiliar with this operating system, which may form a core of the target user group.

This is certainly a valid concern. However, we had multiple interactions with Syntalos users on GitHub – these colleagues did not indicate any problems with setting up Linux, as long as their IT department permitted it. Please note that reviewer #3 sees the fact that Syntalos runs on Linux as a big advantage, because this system opens much broader usage scenarios. In addition, Windows itself contains a Linux compatibility layer which can be easily used to get familiar with the operating system before installing it on hardware for real (we hint to this possibility on page 13). We do believe that it is generally possible to port Syntalos to Windows. However, implementing this option with all features of Syntalos would require multiple adaptations. It would also require a very solid, extensive validation of all functions on Windows which is beyond the scope of the initial description and release of the system.

2) Multiple methods, or “synchroniser constructs”, were used for aligning clocks across modules. If each method is as robust as the other, why not choose a single method to synchronise the data, e.g. “write the offset information to a special tsync binary file”? The authors approach seems unnecessarily complex, which could increase the likely hood of errors/bugs.

We thank the reviewer for his/her question which shows that our text was unclear in this respect. We have now clarified the reasons for our approach in the text. We also added a better general explanation of synchronizers (page 7). In brief, different synchronizer constructs are needed to

account for the different ways in which timing information is provided by different hardware devices. The two synchronizers provided by Syntalos cover a wide variety of different hardware (virtually we could say “all”). They produce synchronized timing information for all active devices in the experiment. In some case, this information can not be stored in the target data format of a given device (e.g. a video file or Intan RHD file). Time-sync files (tsync files) are written for these cases, and allow Syntalos to use common formats for device-specific data storage. The ‘objective’ (synchronized) time information is then loaded from the tsync file.

Reviewer 2:

In this manuscript the authors describe a Linux based software for synchronization of data acquisition systems, called Syntalos. It does this by checking and aligning the acquisition timestamps to a master clock from the Linux computer. It also allows to control actuators within 30-50ms.

The technical details of the system as well as the benchmarking tests to assess synchronization stability are very well described. The authors present one experiment from one mouse in a maze recording calcium imaging and video as application case. Another example of electrophysiological recordings is used to illustrate the risk of misalignment, but it does not seem to have utilized the system.

The authors indicate that Syntalos could be used for “multiple experimental approaches” for “real time closed experiments”. In the abstract, it is mentioned that “an arbitrary number of sources, including multi-channel electrophysiological recordings and different live imaging devices, as well as closed-loop, real-time interventions with different actuators.” However, the electrophysiological equipment is only used during benchmarking and not in a real experiment (unless I misunderstood, and it was used in the Fig. 1 Ephys experiment) and the closed loop described has a ~30ms and not real time response.

The application case presented is well described, but it is also very specific to the problem in hand (e.g., in house designed actuators). For the system to be widely adopted by the neuroscience community, additional examples and a broader range of tested sources and actuators is important.

We thank the reviewer for his/her precise description and generally positive evaluation of our system. We will address the specific points of the reviewer below.

More specifically:

1) To ensure that the title reflects the nature of the system I would suggest adding “synchronization”. For example: “Syntalos: A software for precise synchronization of simultaneous multi-modal.....”

We thank the reviewer for the suggestion and have modified the title of the manuscript. It now reads: ‘Syntalos: A software for precise simultaneous multi-modal data acquisition and closed-loop interventions’.

2) Similarly, in the abstract, it should be made clearer that Syntalos is a software to synchronize data. Please also indicate that the experiments were performed with mice. This might not be obvious outside the neuroscience community.

We agree, and have included the requested changes in the abstract.

3) Closed-loop. As the authors indicate, ~30ms latency is too high for many closed-loop interventions. It is also mentioned that lower latency could be achieved with C++ modules removing the overhead of using Python modules. What latency could be achieved? This is not established. Could you provide benchmarking information for such a closed-loop system? (or even better, an experimental example). One idea would be to repeat the example shown in Fig.5.D with a C++ implementation to compare the two approaches.

Indeed, a 30 ms latency is way too high, given our aim of optimizing temporal precision throughout the experiments. The same point was raised by reviewers #1 and #3 and has prompted us to improve this feature by re-writing parts of Syntalos.

Syntalos' message-passing engine for modules that run as external processes (like the Python module in our example) is now built upon Iceoryx (<https://iceoryx.io/latest/>). This system is also used in cars and robotic automation to enable individual processes to communicate on the same device. Furthermore, many other optimizations have been applied to improve latencies. The roundtrip latency for a Python script running in Syntalos and communicating with an Arduino Uno via the Firmata protocol is now down to 2-6 ms (shown in Fig. 5). The remaining latency is primarily due to the Arduino itself and its connection to the computer (the USB connection confers significant delays when the computer is busy by recording data from multiple devices; see Fig. 3). We note that the Arduino Uno can not communicate via USB and read data at the same time, inducing significant latency in closed-loop applications. Using just an Arduino Uno on its own, others have measured an average roundtrip latency of ~3.16 ms as well (see <https://github.com/seeraven/arduinoLatency>). For devices communicating with the computer via USB, using a real-time Linux kernel would improve the USB-induced latency and time-variance as well. Specialized external hardware can have significantly lower latencies, as we have demonstrated with the Raspberry Pi Pico which Syntalos controls with MicroPython. For this device, we now measured latencies of 50-150 μ s (the new data are shown in Fig. 5).

To make it easy to reproduce our benchmarking results, we provide hardware designs and code to build a precise pulse generator using a cheap Raspberry Pi Pico at <https://syntalos.org/docs/timesync-verification/>. A Syntalos configuration file for the latency test is also provided on the same page. Please keep in mind that the latency tests described in our manuscript were all conducted while Syntalos was recording data from a large array of cameras, Miniscopes and electrophysiology hardware, in order to get a worst-case and realistic scenario (see Fig. 3). Performance will be even better with less USB devices connected to the computer.

We thank the reviewer for their suggestion to write the latency test module purely in C++. This test was performed, but did not substantially decrease the latency. This result can be explained with the USB communication and the limitations of the low-performance Arduino Uno itself, as discussed above. Internal latencies of messages passed between Syntalos modules have been measured in the nanosecond range for internal, threaded modules (extremely dependent on system load) and at around ~10 microseconds for messages passed to modules that run in a separate process.

If an extremely low latency is required, Syntalos can control dedicated hardware to perform this task (as shown in newly added Fig. S4 using as Raspberry Pi 4 and a custom interface board).

Here, the dedicated hardware performs data acquisition, digital filtering, and spike detection, and the latency mainly depends on the signal length that has to be evaluated by the digital filter (31 samples, i.e. 1 ms @ 30 kSamples/s for a band-pass filter with 0.7-5 kHz). For lower frequencies or frequency contents (e.g. ripple oscillations, ~200 Hz, i.e. 5 ms per cycle, several cycles required to detect a ripple complex), the latency will be correspondingly longer.

Again, we thank the reviewers for emphasizing this important issue. With the improved delay times, we are sure that our system allows for closed-loop interventions at any experimentally required speed.

4) It is my understanding that the initial example on mice touching apertures of varying widths was not acquired using Syntalos. How was it acquired and synchronized? Fig 1.C and 1.D seem to suggest that the paradigm was implemented on Syntalos. If it was it would be an interesting example and comparison to the original one. If it was not, this or another Ephys experiment would be interesting.

This impression arose from an unclear section of our original text. We thank the reviewer for highlighting it. Indeed, all experiments and data shown in the paper have been acquired and controlled using Syntalos. This includes all experiments shown in Figure 1, as now clarified in the text (page 4). Data from this experiment are part of a full paper, which is currently under revision and is available as preprint (Heimburg et al. “A tactile discrimination task to study neuronal dynamics in freely-moving mice”; <https://www.biorxiv.org/content/10.1101/2024.08.24.609326v1>). This paper provides, up to now, the most extensive and complete application of Syntalos. We therefore included the reference into our manuscript (page 13).

As requested by the reviewer, we also include a further, different application case where Syntalos acquired data from a combination of in vivo electrophysiology (field potentials) and respiration-induced pressure changed in a plethysmograph (supplementary figure S7 and text page 13).

5) Additional application cases. The abstract mentions that “preliminary experiments with different research questions...” But only 1 application case is shown. Please add more. In line with the above, the authors also mention that they obtained feedback from different laboratories, which allows me to assume that they could provide a larger number of application cases. Presenting more than 1 example and a comparison with other systems would demonstrate the usability and increase the adoption of Syntalos.

We agree that application cases are a core element of a methodological paper like ours. Indeed, all data shown in the present manuscript has been recorded using Syntalos (with exception, of course, of the comparative data from Bonsai). These experiments include (i) unit recordings in a tactile discrimination task (Fig. 1), (ii) Miniscope data in a spatial memory task (Fig. 6) and (iii) field potential and respiration recordings in a plethysmograph (Fig. S7). In addition, technical tests (with recordings of real data) were performed in multiple configurations, including (i) a 24-hour recording from four video cameras, one UCLA Miniscope, one Intan RHD 2000 amplifier, an Arduino Uno and a Raspberry Pi Pico (see Fig. 3-5) and (ii) extensive tests for fast closed-loop applications, as typically needed in optogenetic setups (now reported, with strongly reduced

delays, on page 9-10, page 13 and in Figures 5 and S4; see our response major-(2) to reviewer 1). Finally, Heimburg et al. have now pre-published their full paper ‘A tactile discrimination task to study neuronal dynamics in freely-moving mice’ as a pre-print on bioRxiv (<https://www.biorxiv.org/content/10.1101/2024.08.24.609326v1>). This work was done using Syntalos for data acquisition and closed-loop interactions and is referred to on page 13.

Comparison to existing systems

6.1) In the introduction other software (Bonsai RX, ANY-Maze or Noldus EthoVision XT) is listed and a list of key requirements indicated. It would help to have references for each software and to indicate which of the key requirements each one covers and which ones they do not. This will help understand what Syntalos brings that is different to the others.

We agree that a comparison between Syntalos and existing systems is important. This has also been pointed out by the other reviewers. We therefore provide a standardized short description of commonly used DAQs in the supplementary material (Table S4). We chose an extended table format with short explications of each listed parameter. For example, none of the listed programs have a synchronization method like ours, but they may provide means to synchronize their data by external LED light pulses or alike. Our extended table format does, in our eyes, respect the strengths and limitations of each program without any unfair bias.

In addition, we performed an extensive comparative test by recording multi-modal data with Bonsai. The results are shown in figure S5 and do again reveal several advantages of Syntalos (reliable timestamping, precise synchronization over long recording times and without external cues, ability to record from large numbers of devices). We hope that our comparison does not devalue the specific abilities and advantages of Bonsai which is certainly a good solution for the behavioral neurosciences.

6.2) In addition, in the supplementary data there is a comparison to some of them which could be moved to the discussion.

Due to space constraints, we could not move the whole comparison into the discussion. However, we now added more detail to the discussion, as requested by the reviewer (pages 12). In addition, we have expanded the comparison in the supplementary information which does now contain Simulink, OpenEphys GUI and LabVIEW (supplementary Table S4).

In essence, our comparison underlines that Syntalos provides a uniquely versatile, easy-to-use and precise system for synchronization of multi-modal data and closed-loop operations in complex experimental setups.

6.3) Furthermore, there is no comparison to other systems commonly used in neuroscience such as OpenEphys (e.g., with the video frame grabber plugin or Bonsai). Please add some additional comparisons (e.g. to Tarcsay et al., eNeuro 2022 and Buccino et al., J Neural Eng 2018).

We have added a comparison with the OpenEphys GUI to the list of systems described in the supplementary material (Table S4). We also included the two references suggested by the reviewer and report that Tarcsay et al.(2022) and Buccino et al. (2018) both used Bonsai, and that Buccino et al. (2018) combined it with an extension to the OpenEphys GUI (page 10 and

12). For a practical test and comparison of recording systems we restrained our revision to the open-source program Bonsai, not at least due to the high costs of the required hardware to run the OpenEphys GUI. Nevertheless, we hope that table S4 gives a reasonable account of the performance of both systems. Together with our practical, data-based comparison between Bonsai and Syntalos we hope that we provide a balanced and informative impression of Syntalos' performance.

6.4) How different it is from a Simulink Real-Time or Labview implementation? These in principle would also allow to impose a fix timestamp and also provide a block diagram implementation.

LabVIEW and Simulink are both visual programming languages, while Syntalos is not. While the former two allow expressing turing-complete programs in visual blocks, Syntalos defers to text-based programming languages like Python for that task. Syntalos is best comparable to the workflow of a digital audio workstation, and its modules resemble full devices that can be connected with wires just like physical devices would be.

LabVIEW works best with devices provided by National Instruments, while Syntalos is 'device-agnostic' and thereby most versatile. Both LabVIEW and Simulink do not perform the same kind of dynamic synchronization that Syntalos implements. Both solutions are also not open source.

In summary, we respectfully defend our view that Syntalos is unique in its versatility and its temporal precision (synchrony) for running experiments with multiple devices and data formats. We have clarified this on pages 3 of the revised manuscript, page 6 in the Supplemental Information, and in our systematic comparison in Table S4 (we hope, we did so in respectful terms acknowledging the merits of alternative approaches).

6.5) Ideally a direct comparison with one or some of these systems (most interestingly with Bonsai) would help understand the advantage of using Syntalos.

Indeed, we performed extensive experimental tests with Bonsai and do now provide direct comparative data. A new figure and an extensive description have been added to the manuscript (Fig. S5 and page 10 and 12).

System capabilities and characteristics.

7.1) Other devices. Could you indicate more clearly what the process is for getting timing information from devices without a clock signal?

Thank you - the respective section has been clarified in the manuscript (page 7). For devices that provide no timestamps and are strictly polled instruments, Syntalos takes the mean of the master clock's time before request for data and response. For devices with buffers, Syntalos calculates the apparent recording time backwards, using the time of data receipt and the filling state of the buffer.

7.2) Could it record from multi subjects? How would this be achieved? (e.g. Kimchi et al., eNeuro 2020).

Yes, such an experiment can be easily done with Syntalos as long as the used hardware devices do not conflict with each other. Syntalos can handle an arbitrary amount of inputs, being only

limited by the computer hardware. As an example for a setup with multiple data acquisition sources, we recorded data from 4 cameras, 1 Miniscope, 1 Intan-based headstage and a sensor driven by a Raspberry Pi Pico simultaneously without any issues (see page 8 of the revised manuscript).

OpBox (Kimchi *et al.*, *eNeuro* 2020) is an extremely interesting project, especially from a hardware perspective. There should be no problem to write a Syntalos module for it that acquires data from multiple subjects. OpBox is a combined hardware and software solution, of which Syntalos could certainly fulfill the DAQ software part.

7.3) What would happen if a device were disconnected halfway through the experiment? Would it keep running? Would the other modules stay connected?

This is an interesting question, thank you. The intended design of Syntalos is that, in such cases, the entire experimental run is stopped and all data which have been up to that point is written to disk. This would happen if only one module encounters an error. It ensures that no experiments are recorded with incomplete data, and it avoids that a module failure goes unnoticed.

However, due to the reviewer's comment, we added a new feature to Syntalos which allows the user to explicitly disable the "stop-on-failure" behavior for each module individually. In that way, modules can be marked as "optional" if they are not essential and failure is allowed. Upon encountering an error, the optional module will indicate the error by switching to its "Failed" state (turning red and displaying the "Broken" icon in Syntalos' GUI). The error messages of failed modules will still be displayed once the user manually stops the experiment run. The run will be marked as a success, but all error messages of failed modules will also still be written into the experiment's EDL metadata, for reference. This new feature is now described on page 6.

7.4) Line 344 says that "performance could also be visualized immediately and tracked in real-time". How?

We thank the reviewer for his/her hint - this phrase has been adjusted in the text. Syntalos contains a dedicated plotting module that can plot the requested performance graph in real time. Furthermore, using a Python module, animal movements can be drawn into the video frames and displayed as well (now explained on page 11).

7.5) EDL format. Please compare to standard formats. Why is this data structure selected instead of already implemented formats in neuroscience?

Indeed, such a comparison is important, given the plethora of data formats and the difficulties of compatibility. We have now compared EDL with other commonly used formats at the EDL website: <https://edl.readthedocs.io/latest/intro.html#comparison-with-related-projects>. We also added a respective section to our supplement (pages 3-4). The main advantage of using EDL for Syntalos is that it supports the specific requirements of recording from many different data sources in real-time. This can not be done by any HDF5-based format at the same performance level. Furthermore, Syntalos' specific features make it impractical to use formats that strictly map to the HDF5 abstract data model – this would limit the data types that Syntalos can use (such as is the case with Exdir). Therefore, we do believe that EDL is both a capable (due to its performance advantages in live recordings) as well as a practical solution. It enables reusing

many other commonly used formats, such as Pandas JSON tables, Intan's RHD format, or video files, which simplifies its adoption by other groups and its integration into existing data analysis pipelines.

Additional minor suggestions:

8.1) A couple of videos (could be screen recordings) demonstrating how to use Syntalos would be helpful.

We added video tutorials demonstrating some of the features of Syntalos and giving users an easier start. They can be found alongside the written tutorials on the Syntalos website at:

<https://syntalos.org/tutorials/>

8.2) There are no references in Figure 4 (I understand that they are the same as in Fig 3 – but please add them).

Sorry. The respective reference / legend has been added.

8.3) Fig 6. A diagram of Syntalos in this experiment would help understand how to use it.

We have added the respective connection diagram of Syntalos' modules as well as a logic diagram of the actions take in a forced M-maze trial in a new supplementary Fig. S6.

8.4) Lines 100-107 seem to be part of the general architecture.

Thank you – we moved the subtitle to the correct place.

Remarks on code availability:

I could not install the software on my available Linux machine (the required operating system), but I looked at the code and it seems well structured and documented.

We have added a quick-install description for users in addition to the more detailed installation instructions to the (also completely rewritten and new) Syntalos website: <https://syntalos.org/get/>

For Ubuntu Linux users, we have also created a PPA to easily install and update the software. We gave the software to 5 users with the newly added instructions, 2 of them not even running Linux yet, and they were able to install and run it without additional questions. Thus, we hope that with these additions installation is now much easier.

Reviewer 3:

Modern neuroscience requires progressively more complex experimental setups that incorporate multiple modalities for recording or manipulating neurons while capturing and controlling animal behaviour with sub-second time resolution. Such complex and custom-made setups would benefit greatly from using a software tool that combines multimodal data acquisition and control, ensuring a high degree of synchrony between data sources. Klumpp et al. present a Linux-based open-source software solution for this, called Syntalos, which offers an easy-to-use interface that allows highly precise and synchronous multimodal recordings and closed-loop device interactions. The software presented serves an important niche as most competing software relies on the restrictive operating system environment of Microsoft Windows, while Syntalos is Linux-based. Also, as our lab and those of colleagues' have experienced time and again, multimodal data synchrony is key to more and more experiments but often difficult to achieve when operating custom-made setups consisting of diverse components. This is an exciting manuscript presenting a software tool that will enable a growing number of labs to record and control such multimodal experiments, while ensuring synchrony between data modalities. The figures and text provided by the authors largely allows a non-expert audience to understand the rationale behind Syntalos, how it works, and what it achieves. I believe this manuscript is, overall, well suited for publication in Nature Communications, but I have a few suggestions that would increase the impact of both the manuscript and the accompanying software:

We thank the reviewer for his/her positive and encouraging comments general assessment of Syntalos.

Major:

1.1) Syntalos is, of course, not the only software solution for multimodal data acquisition and closed-loop experimental control. The authors rightly acknowledge the widely used free tool Bonsai RX and commercial solutions ANY-Maze/EthoVision. I appreciate the gap Syntalos is filling here in both providing a Linux version (all other software solutions work on Windows only) and investing considerable effort in establishing a user-friendly interface with a high degree of synchrony between data sources. Both specificity and added value should be made clearer though. I suggest incorporating "Syntalos: A Linux-based software..." into the title so that potential users can easily find it and die-hard Windows users will know this won't work on their system right away.

Again, we thank the reviewer for highlighting what we believe are the stand-alone features of Syntalos: its high versatility in integrating multiple data sources, its user-friendly design, and the precise long-term synchronization of devices. In addition, we agree that running it under Linux is an advantage, but also might detain some Windows users from testing it. From our own experience and our interactions with different colleagues, installing Linux on a measurement system has been no issue even for rather inexperienced users. We also note (page 13 of the revised manuscript) that Windows users can explore the Syntalos GUI by running it on Microsoft's Linux compatibility layer (Windows subsystem for Linux, WSL2), though with degraded performance. We do provide detailed instructions for this possibility at <https://syntalos.org/docs/setup/install-windows/>.

We acknowledge the reviewer's suggestion and intention to make Linux already visible in the title. However, we feel that the title might become overly technical with this addition, and therefore decided to add 'Linux-based' to the initial sentences of the abstract. We have also included it into the list of keywords such that it is highly visible for all potential readers. We hope that this meets the intentions of the reviewer. We note that, in the near future, it might become possible to port the software to Windows (see also our comment *minor-(1)* to reviewer 1).

1.2) Furthermore, as with any piece of software that has comparable competitors, I strongly suggest the authors quantify their comparisons instead of just drawing hypothetical comparisons. Specifically, I suggest the authors compare multimodal synchrony across the 4 different software tools. In addition, a table listing the features Syntalos provides and the OS it operates on besides those of the other competing pieces of software would be beneficial.

This concern has been an issue for all three reviewers, and we agree that it should be addressed more explicitly. We did so by performing extensive test runs with Bonsai, and we present the results on pages 10 and 12 of the main manuscript, in supplemental figure S5 and in the supplemental table S4. In line with our responses to reviewer 1 (major-1) and reviewer 2 (6), we would like to point out the following arguments:

- Many of the competing solutions are proprietary and designed for specific hardware (Plexon, ANY-maze, etc.). This is a severe scientific and financial restriction for many labs. It also made it difficult to perform extensive tests with such systems (e.g., testing the comparatively cheap and open-source OpenEphys ONIX system would have prompted investments of around € 13,000 for the required hardware).
- Since most published open-source DAQ systems use Bonsai in some capacity, we focused on comparing Syntalos to Bonsai. The results are now documented in figures 3-5 and S5 and on pages 10 and 12. In brief, they reveal several advantages of Syntalos (reliable time-stamping, precise synchronization over long recording times and without external cues, ability to record from large numbers of devices). Nevertheless, we acknowledge the specific strengths of Bonsai and hope that our comparison does not devalue its advantages.
- We include a systematic comparison between Syntalos and many of the most widely used systems in Table S4. We acknowledge, however, that different DAQ systems fill different niches, such that a highly standardized comparison in a table would not do justice to their individual advantages. For example, some systems like LabVIEW are primarily visual programming languages as commonly used for instrument control, while a solution like ANY-maze emphasizes video tracking and was specifically designed for animal experiments, with a very different programming model. Such subtle differences need to be accounted for. Our best attempt to provide a reasonable comparison is now a list of key features for each of the programs.

As a conclusion, we believe that Syntalos does provide high versatility, very good usability, and exceptional temporal precision across all devices for a large variety of possible experimental designs. In this respect, we feel that Syntalos is unique and novel, and that it meets a growing

and urgent need for hardware integration, software support and objective timing in complex neuro-behavioral settings with multi-dimensional data acquisition.

2) I commend the authors for establishing a systematic experiment directory layout (EDL), but it is not clear to me what the benefits of EDL are compared to the more and more widely used Neurodata Without Borders (NWB) format that aims to unite neurophysiological and behavioural data across labs. I believe Syntalos and the wider open data community would benefit greatly if Syntalos (also) offered to write and read data in the NWB format, unless I'm missing something here (in which case, this should also be pointed out in the text). Given that the EDL format has been developed in-house and has no associated publications yet as far as I'm aware, some introduction to EDL and why it is being used here should also be provided.

We thank the reviewer for this hint which refers to an issue raised by Reviewer 2 as well. We now explained the rationale for the development of EDL in more detail on page 2-4 of the Supplemental Information and on the EDL specification website at <https://edl.readthedocs.io/latest/intro.html> (this page also includes a written comparison of EDL with other data formats that we have included in our supplementary material, see 'Comparison of the EDL data storage layout to other data formats' in the supplementary information). The main reasons to use EDL are (1) to overcome the limitations of the HDF5-backed NWB format for live recordings and to allow the software to write out incoming data as fast as possible, and (2) to have a format expressive enough to contain all metadata that Syntalos generates. Inside of an EDL directory structure, well-known formats (such as video files and Intan's RHD file format) are used to make it easy for experimenters to incorporate their data into existing data processing pipelines.

For the needs of individual laboratories, the EDL data can be converted into any desired format. This permits groups to incorporate the specific data required for their analysis in the appropriate formats for their analysis tools. For example, a group in our department developed a converter to transform Syntalos-generated EDL data into NWB files for use with SpikeInterface for electrophysiology: <https://github.com/catalystneuro/mease-lab-to-nwb>. Discussions on Syntalos' GitHub page also made us aware of a lab that has written a tool to convert some of Syntalos' EDL data into a format suitable for MoSeq2 (<https://dattalab.github.io/moseq2-website/>).

In summary, we believe that the EDL format provides key advantages for live recording of data, while not making it hard for researchers to use the recorded data in any way they need, including conversion into other formats post-hoc. These options are now summarized more clearly and explicitly on page 3-4 of the Supplemental Information.

3) Syntalos has a potentially very broad appeal to many labs performing in vivo neurophysiology experiments based on its various modules, yet the authors show only data of a (de-)synchronised electrophysiology/behaviour experiment (Figure 1A-C) and a miniscope/camera-based locomotion experiment (Figure 6). Given the prevalence of (1) two-photon / head-fixed camera / analogue behaviour recordings and (2) optogenetic stimulation / freely moving behaviour, I suggest the authors include at least one further set of one or both modalities into their data, showing the respective Syntalos configuration required, and how the acquired data looks, and how desynchronisation can be ruled out.

Please allow us to reiterate parts of our response to reviewer 2 who had a similar concern. Our manuscript includes original data from three different experiments: (i) unit recordings in a tactile discrimination task (Fig. 1), (ii) miniscope data in a spatial memory task (Fig. 6) and (iii) field potential and respiration recordings in a plethysmograph (Fig. S7). In addition, technical tests (with recordings of real data) were performed in multiple configurations, including (i) a 24-hour recording from four video cameras, one miniscope, one Intan RHD 2000 amplifier, an Arduino Uno and a Raspberry Pi Pico (see figures 3-5) and (ii) extensive tests for fast closed-loop applications, as typically needed in optogenetic setups (now reported, with strongly reduced delays, on page 9-10 and in figure 5; see our response major-(2) to reviewer 1). Finally, Heimburg et al. have pre-published a full paper using Syntalos and reporting strictly timed correlations of perceptual tasks and single-unit activity: ‘A tactile discrimination task to study neuronal dynamics in freely-moving mice’ as a pre-print on bioRxiv (<https://www.biorxiv.org/content/10.1101/2024.08.24.609326v1>). We refer to this paper on page 13. Unfortunately, neither we nor any of our cooperation partners has a running license for interventional animal experiments as requested by the reviewer. It was impossible to get a respective extension of licenses within the available time for correction of the paper. We hope, however, that the evidence from the above-mentioned experiments and tests is an adequate proof-of-principle for the functionality and reliable timing control of our system.

Minor:

4) *Although not strictly part of the manuscript, I believe the software would find wider acceptance if the authors incorporated further improvements into their documentation (<https://syntalos.readthedocs.io/>), such as (1) a simplified installation (I had trouble with Flatpak on Ubuntu 22.04 and the package-based installation notes say “After the PPA is registered...” without making mention of the PPA address), (2) releasing video tutorials instead of text/graphic only, and (3) completing some important documentation content, e.g. Introduction “Design Goals: Coming soon! Architecture Overview: Coming soon!”.*

We apologize for having caused these difficulties and have taken various measures to simplify installation and facilitate usage of the software.

(1) In the course of the revision, the documentation has been extended and improved. Instead of the ReadTheDocs-based linear documentation pages, we have registered a website for Syntalos (<https://syntalos.org>) and restructured the existing documentation to make it easier to read. We also expanded the documentation greatly. For installation on Ubuntu, we now provide a PPA with Syntalos precompiled, as well as quick-install instructions for people who do not need the detailed instructions and just want to get started quickly. All this information is linked at <https://syntalos.org/get/>

(2) We have created a set of video tutorials which can be found at <https://syntalos.org/tutorials/> alongside the written tutorials.

(3) The documentation pages should now be complete and have no missing information / placeholders anymore.

5) *I understand the miniscope data in Figure 6 is mostly for demonstration purposes only, but having acquired miniscope data in dCA1 for several years, I am not convinced by the quality of*

data presented. This is hard to judge from the limited amount of actual calcium data presented (some raw calcium traces with position/velocity data should be presented), but the fourth panels from the left in 6B and 6D suggest a strongly oblique field of view on dCA1 that would allow the capture of only a comparatively small number of neurons. As a point of reference, I suggest the authors inspect data and protocols from Yaniv Ziv's recent work (Geva et al., 2023, Neuron, <https://doi.org/10.1016/j.neuron.2023.05.005>, e.g., Figure 2C).

Thank you for hinting towards this issue. The presented data have been gathered in a special series of experiments aiming at post-hoc identification and detailed histological analysis of the recorded cells. For this, we intentionally used sparse-labeling of pyramidal neurons by heavily diluting the Cre virus activating GCaMP-expression in the lowest possible number of CA1 pyramidal cells, ideally excluding neighboring cells. Thus, the experiment was not optimized for miniscope data from multiple cells, but rather for sparse labelling of few, identifiable cells. We have now pointed out this limitation on page 10 and refer to the paper from the Ziv group. We also added raw traces and velocity information to Figure 6, making the experimental approach more understandable (see the next point below).

6) It is not entirely clear to me how Syntalos performs mouse tracking as for Figure 6, as depicted in Figure 1D to the left. Is this based on a DeepLabCut module? Does it require a pretrained DLC model? Or is it based on background contrasting? Does it allow online device control? The authors should make clear how this works to boost readers' understanding how it could be useful for them.

In these experiments, Syntalos used data from custom-built light barriers which were integrated into the M-maze to determine the mouse's position during the experiment. The light barrier status was read using the Arduino, via Syntalos. After the experiment, video data was analyzed using DeepLabCut, with a model which was trained specifically on M-maze data to detect the mice even with Miniscopes attached.

Syntalos is able to perform DeepLabCut detection online using DLC Live. However, this method was not used for the experiment depicted in Figure 6. The experiment is now explained in some more detail on pages 10f and in figures 6 and S6.

7) I have come across some typographic mistakes: (line 35 "Jensen", Fig 2 "Syntlos", Fig 5 "guarantied").

These mistakes have been fixed, thank you for noticing and mentioning them!

8) The full name for "IPC" (mentioned in Figure 2) should be stated.

This has been changed.

9) Fig 2B: It is unclear what "Engine" and "Module" explicitly refer to at top / bottom. Could the authors clarify how this figure is set up?

This has now been clarified explicitly in the figure's description.

10) Could the authors provide a table/list which types of cameras are currently supported? Our lab and those of colleagues largely rely on Basler cameras for example. Are these supported?

Thank you for this question. Syntalos now supports any industrial camera that supports the GenICam® standard via a newly created module that utilizes the Aravis implementation of the GenICam specification. It also supports any consumer camera that supports the UVC Video standard for USB devices. We also are aware of Syntalos users who are working on support for 3D vision cameras.

We obtained a Basler camera and tested it successfully using the new Aravis Camera module of Syntalos. The data from it is included in the updated Figures 3-5.

GenICam is supported by nearly all industrial cameras created in the last decade, and most camera manufacturers are even members of the GenICam workgroup.

We added a small table stating this fact and the tested devices to our supplementary material in Table S5.

11). Could you please improve either the way Figure 4E right panel displays the data or how it is described? As it stands, I could not grasp what is depicted, unfortunately.

Thank you for the hint. We have adjusted the description to make the purpose of this panel more clear.

Reviewer #4:

Remarks to the Author:

I co-reviewed this manuscript with one of the reviewers who provided the listed reports as part of the Nature Communications initiative to facilitate training in peer review and appropriate recognition for co-reviewers.

Remarks on code availability:

I did not have access to a Linux machine to run the software but was able to confirm access to the github database which had code available for download and a corresponding flathub page which aids in distribution of Linux software.